# ANTIGENLM: STRUCTURE-AWARE DNA LANGUAGE MODELING FOR INFLUENZA

**Yue Pei[1,4], Xuebin Chi[1,4],\* Yu Kang[2,3,4]\***

[1]Computer Network Information Center, Chinese Academy of Sciences
[2]Beijing Institute of Genomics, Chinese Academy of Sciences
[3]China National Center for Bioinformation
[4]University of Chinese Academy of Sciences
`ypei@cnic.cn, chi@sccas.cn, kangy@big.ac.cn`

## ABSTRACT

Language models have advanced sequence analysis, yet DNA foundation models often lag behind task-specific methods for unclear reasons. We present AntigenLM, a generative DNA language model pretrained on influenza genomes with intact, aligned functional units. This structure-aware pretraining enables AntigenLM to capture evolutionary constraints and generalize across tasks. Fine-tuned on time-series hemagglutinin (HA) and neuraminidase (NA) sequences, AntigenLM accurately forecasts future antigenic variants across regions and subtypes, including those unseen during training, outperforming phylogenetic and evolution-based models. It also achieves near-perfect subtype classification. Ablation studies show that disrupting genomic structure through fragmentation or shuffling severely degrades performance, revealing the importance of preserving functional-unit integrity in DNA language modeling. AntigenLM thus provides both a powerful framework for antigen evolution prediction and a general principle for building biologically grounded DNA foundation models.

## 1 INTRODUCTION

Influenza viruses evolve rapidly to escape host immunity, driving seasonal epidemics and necessitating frequent vaccine updates (Han et al., 2023). Accurate forecasting of viral evolution is therefore essential for vaccine strain selection and for mitigating global public health burden (Matz & Ellebedy, 2025). Current vaccine recommendations integrate large-scale genomic surveillance, antigenic characterization, and epidemiological monitoring, coordinated by the World Health Organization (WHO) (Bucholc et al., 2025).

Traditional forecasting approaches rely on phylogenetic tree dynamics and mutation-based predictive models (Hadfield et al., 2018; Huddleston et al., 2020; Łuksza & Lässig, 2014; Neher et al., 2014). More recently, deep learning models (Mehrotra et al., 2025; Lou et al., 2024) have been shown to accurately predict clade-level mutation trajectories. *beth-1*, which explicitly models sitewise substitution dynamics, improves genetic matching relative to tree-based methods and underscores the promise of machine learning for vaccine guidance(Lou et al., 2024). In parallel, genomic foundation models (e.g., DNABERT (Zhou et al., 2024), NT (Dalla-Torre et al., 2024), GROVER (Sanabria et al., 2024), HyenaDNA (Nguyen et al., 2023)) and protein foundation models (e.g., ESM(Rives et al., 2021), ProtGPT2(Ferruz et al., 2022)), together with influenza-focused protein predictors (Hie et al., 2021; Ma et al., 2024; Ito et al., 2025), offer another powerful approach for antigenic sequence representation.

However, viral evolution is shaped by coordinated interactions across the entire genome (Cobey, 2024; Gouma et al., 2020). These include RNA–RNA interactions and co-packaging (Bolte et al., 2019; Yang et al., 2024), constraints on segment reassortment (Holmes, 2007), and co-adaptation between polymerase segments (PB1/PB2/PA) and antigenic proteins HA and NA (Vigeveno et al., 2024; Noda, 2020) . Models that ignore this structural context—such as site-wise predictors like

---

*Corresponding authors.

*beth-1*—fragment biological signals, limiting both generalization and interpretability. General-purpose foundation models, trained on heterogeneous multi-species genomes, tend to capture local sequence patterns but often fail to model these genome-wide, higher-order constraints, resulting in degraded performance compared to species-aware and structurally aligned pretraining (Karollus et al., 2024; Benegas et al., 2025).

Moreover, although protein-level forecasting can capture aspects of antigenic evolution, key determinants of viral fitness—including synonymous mutations, noncoding regulatory elements, RNA secondary structures, packaging signals, codon-usage–mediated host adaptation, and polymerase compatibility—are completely invisible to protein-only models. Extensive experimental evidence highlights the importance of these nucleotide-level mechanisms(Canale et al., 2018; Kryazhimskiy et al., 2008; Fujii et al., 2005; Liu et al., 2025; Gu et al., 2019), and recent benchmarks show that DNA-based models outperform protein-based approaches on related predictive tasks (Boshar et al., 2024). Together, these observations motivate an influenza-specific, whole-genome, nucleotide-level language model for accurate antigen sequence prediction. The compact influenza genome ($\sim$13 kilo-nucleotide) further makes such a specialized model feasible.

Here, we introduce AntigenLM, a generative DNA language model that explicitly preserves segment order and functional-unit integrity during pretraining. The autoregressive model is then finetuned to forecast the antigen sequences of upcoming dominant strains using serial HA–NA sequences collected in fixed temporal windows, which implicitly encode evolutionary trajectories. By enforcing segment orientation and antigen boundaries, AntigenLM learns representations that integrate local sequence dependencies with global genomic structure, enabling generalization across clades, subtypes, and geographic regions.

We evaluate AntigenLM across diverse influenza forecasting tasks and benchmark it against general-purpose foundation models, the current WHO selection method, and state-of-the-art evolutionary predictors such as *beth-1*. We further conduct ablation studies using antigen-only, truncated-genome, and segment-shuffled variants to isolate the contributions of whole-genome context and functional-unit preservation.

Our contributions are threefold: 1. Functional-unit–aware pretraining: We introduce a DNA language model that enforces the integrity and correct permutation of functional units during pretraining and demonstrate its advantage through controlled ablations. 2. Improved influenza forecasting: AntigenLM outperforms state-of-the-art site-based evolutionary models, achieving lower amino acid mismatch in antigenic sequence prediction. 3. Generalizable framework: We provide a blueprint for incorporating biological structure into generative models, with implications for predictive genomics, vaccine design, and modeling rapidly evolving pathogens.

## 2 RELATED WORK

### 2.1 CLASSICAL EVOLUTIONARY FORECASTING METHODS

Classical phylogenetic models form the foundation of current WHO vaccine recommendations. Approaches such as Local Branching Index (LBI)–based tree dynamics (Neher et al., 2014; 2016) and hemagglutination-inhibition (HI)–derived antigenic predictors (Du et al., 2017) rank circulating lineages and estimate their likelihood of future dominance. While effective at integrating serological and sequence data, these methods assume relatively homogeneous site dynamics and do not explicitly account for higher-order constraints or genome-wide coordinated evolution.

### 2.2 DEEP LEARNING–BASED EVOLUTIONARY MODELS

Recent deep-learning approaches model influenza evolution directly from viral sequences. *beth-1* (Lou et al., 2024) infers site-wise mutation fitness from genomic sequences and population seropositivity, enabling quantitative prediction of clade-specific fitness landscapes. Despite strong performance, it treats mutations as independent events, limiting its ability to capture coordinated changes spanning HA, NA, and other segments. EVE (Frazer et al., 2021), a deep generative model trained on H1N1 HA sequences, predicts clade-level mutation trajectories (Mehrotra et al., 2025) but focuses solely on a single antigen and cannot generalize to other subtype antigens smoothly.

## 2.3 GENOMIC LANGUAGE MODELS

Nucleotide-level language models such as DNABERT (Ji et al., 2021), NT (Dalla-Torre et al., 2024), and HyenaDNA (Nguyen et al., 2023) learn contextual representations of DNA sequences. Advances in long-context modeling—e.g., genome-scale Transformers (Fishman et al., 2025) and extended-context Hyena architectures—allow processing of sequences up to megabase length, sufficient for viral genomes. However, these general-purpose models are trained on heterogeneous, multi-organism genomic corpora with vastly different genome sizes and architectures, making it difficult to retain organism-specific structural organization or genome-wide higher-order constraints.

## 2.4 GENERAL-PURPOSE AND INFLUENZA-SPECIFIC PROTEIN LANGUAGE MODELS

Protein language models, including general models such as ESM (Rives et al., 2021; Lin et al., 2023) and influenza-focused models (Hie et al., 2021; Ma et al., 2024; Ito et al., 2025), provide strong antigenic sequence representations and have demonstrated utility in antigenic characterization. However, protein-only models cannot capture key nucleotide-level evolutionary mechanisms—such as synonymous substitutions, codon-usage adaptation, or non-coding regulatory elements—that influence viral fitness without altering protein sequences. Moreover, most protein LMs are not autoregressive and cannot generate full-length antigen forecasts except ProtGPT2 (Ferruz et al., 2022).

## 3 METHOD

### 3.1 MODEL OVERVIEW

We present AntigenLM, a Transformer-based(Vaswani et al., 2017) framework tailored for modeling influenza A viral genomes (**Figure 1B**). The model is derived from the GPT-2(Radford et al., 2019) architecture but redesigned to address the unique challenges of biological sequence learning: (i) extremely long input contexts (up to 13k nucleotides per genome(Van den Hoecke et al., 2015)), (ii) multi-task learning objectives that combine generative and discriminative tasks, and (iii) the need to capture dependencies across distinct functional gene segments(Neverov et al., 2015).

The backbone is a decoder-only Transformer with 6 layers, 384 hidden dimensions, and 6 attention heads. Each block uses a feed-forward sublayer with an inner dimension of 1,536 (i.e., 4× the model dimension) and GELU activations, following the GPT-2 implementation. Although compact compared to large NLP models, the architecture is extended with a 13,000-position embedding range, allowing full-genome modeling without truncation and avoiding hand-crafted tokenizers like BPE and k-mer(Li et al., 2024). This strikes a balance between coverage and computational efficiency, enabling training on tens of thousands of viral genomes using standard multi-GPU setups.

On top of the backbone, we implement a dual-head design for multi-task learning:

1. Language Modeling (LM) Head: tied to the embedding matrix, predicts the next nucleotide token in an autoregressive manner, capturing evolutionary dynamics.
2. Classification Head: projects hidden states at sentinel positions into subtype logits, supporting supervised sequence-level discrimination.

The model leverages shared pretraining, optimizing each task independently during fine-tuning. This enables the model to capture global genome-wide context through autoregressive modeling while benefiting from supervised subtype classification, resulting in richer generative representations and improved subtype identification performance.

### 3.2 FUNCTIONAL-UNIT ENCODING

AntigenLM employs a **two-stage functional-unit encoding** strategy to preserve both genome-wide context and segment-level structure. During **pretraining**, all eight influenza A segments (PB2, PB1, PA, HA, NP, NA, MP, NS) are concatenated into a single continuous full-genome sequence (Hoffmann et al., 2000), maintaining a fixed order and approximate positional alignment. This enables the Transformer to capture long-range co-evolutionary dependencies, such as compensatory mutations between HA and NA (Koel et al., 2013), without artificial markers or truncation and

avoids heavy structural tagging (Huddleston et al., 2020). During **fine-tuning**, sentinel tokens (e.g., <HA>, <NA>) explicitly delimit functional regions (Johnson et al., 2017), guiding attention for subtype classification and constraining autoregressive decoding within each antigen to prevent cross-segment continuation (Sennrich et al., 2015). This combination of scalable unsupervised pretraining and task-specific structural supervision produces biologically faithful, task-adaptive representations.

### 3.3 COMPLEXITY AND EFFICIENCY

Scalability is crucial for genome-scale modeling. AntigenLM is therefore designed to be both compact and long-context. The backbone uses only 6 layers with 384 hidden dimensions and 6 attention heads, but supports sequences of up to 13,000 positions, enabling full-genome modeling without truncation. Both generative and discriminative tasks share this single Transformer backbone, which reduces memory usage and promotes efficient transfer of representations(Avsec et al., 2021). In practice, pretraining on more than 54,000 complete influenza A genomes and subsequent task-specific fine-tuning are feasible on standard multi-GPU setups (see Section 4.5 for training details). These choices make AntigenLM suitable for large-scale influenza surveillance data while remaining computationally tractable.

### 3.4 FINE-TUNING FOR FORECASTING AND CLASSIFICATION

We fine-tune AntigenLM on two downstream tasks using the same GPT-2 style backbone and different heads. For generative forecasting, we train the language modeling head on prompts that concatenate three historical HA/NA blocks followed by one future block. Each example has the structure

$$\underbrace{\text{block}^{(1)}}_{\text{past}} \underbrace{\text{block}^{(2)}}_{\text{past}} \underbrace{\text{block}^{(3)}}_{\text{past}} \underbrace{\text{block}^{(\star)}}_{\text{future}},$$

where

$$\text{block}^{(i)} = \texttt{<subtype>} \texttt{<HA>} \text{HA}^{(i)} \texttt{<NA>} \text{NA}^{(i)} \texttt{<sep>}$$

for $i = 1, 2, 3$. Here $\text{HA}^{(i)}$ and $\text{NA}^{(i)}$ are HA/NA nucleotide sequences from three past time points within the same season. The future block $\text{block}^{(*)}$ follows the same pattern but uses HA, NA from the future time point to be forecast and simply omits the leading <subtype> token. We optimize the standard causal language modeling loss over the full token sequence (excluding padding), and at inference time we feed only the three historical blocks and autoregressively generate the future block starting from the target <HA> token until <sep> is produced or a maximum length is reached.

For subtype classification, we use the same backbone with the classification head and a simpler input that contains a single HA/NA pair and a separator,

$$\texttt{<HA>} \text{HA} \texttt{<NA>} \text{NA} \texttt{<sep>}.$$

The sequence is encoded by the Transformer, and we take the hidden state at the final token position as a compact representation of the HA/NA pair. This representation is passed through a linear layer to produce subtype logits and trained with a cross-entropy loss. In practice we fine-tune two task-specific models—one generative and one discriminative—both initialized from the same pretrained AntigenLM backbone, and we also support joint optimization of both objectives via a simple weighted sum of the language modeling and classification losses.

## 4 EXPERIMENTS

### 4.1 DATASETS

We assembled a comprehensive corpus of influenza A virus genomes from GISAID (Shu & McCauley, 2017), including the two major subtypes A/H3N2 and A/H1N1 as well as 10 minor subtypes (e.g., H5N1, H7N9). For both pretraining and fine-tuning, we used sequences collected before February 2022 and applied stringent quality control by removing duplicates, excluding incomplete genomes (fewer than eight segments), and filtering out sequences with $< 90\%$ mean segment coverage relative to the corresponding subtype reference. After filtering, the dataset comprised 32,758 H3N2, 20,680 H1N1, and 1,074 minor-subtype genomes, totaling $\sim$600 million nucleotides across

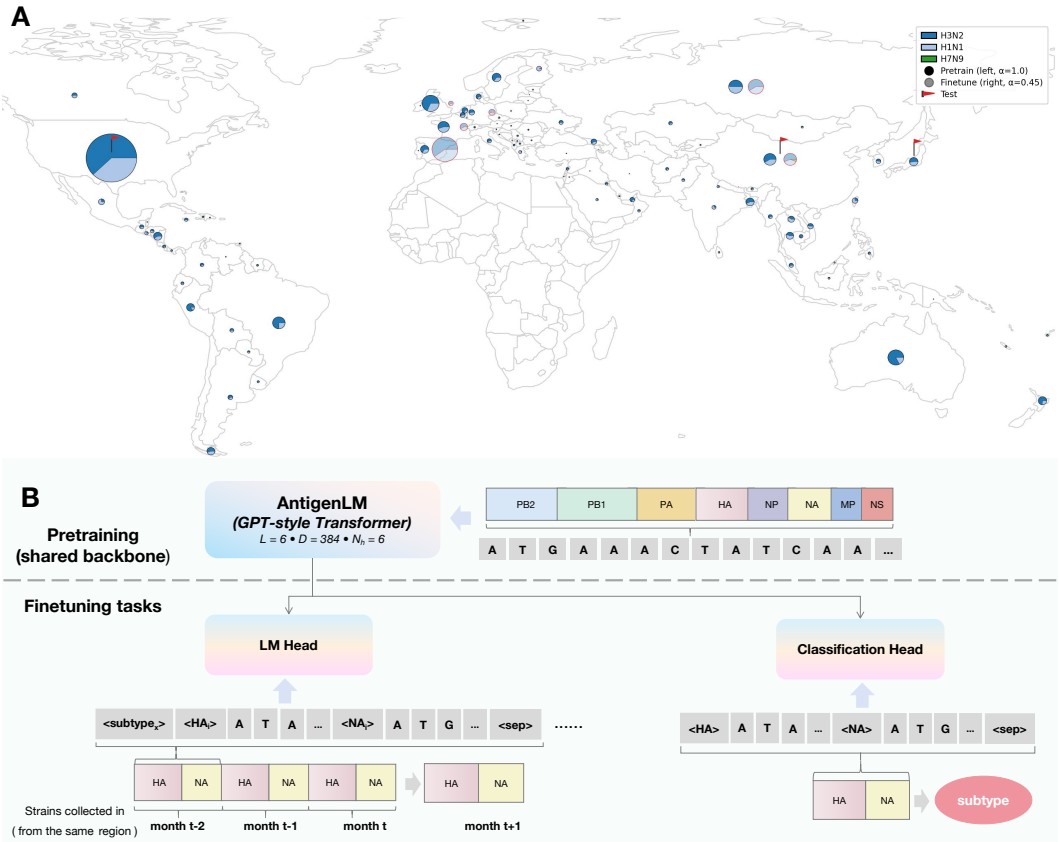

Figure 1: Data Distribution and AntigenLM Architecture. (A) Global distribution of influenza A virus sequences used for pretraining, fine-tuning, and testing. Circle size reflects sample count; pie sectors show subtype composition. Dark circles represent pretraining data, light circles fine-tuning data, and red ticks mark test regions. (B) AntigenLM architecture. Schematic illustration of the pretraining and finetuning phases. The model utilizes a GPT-style Transformer as a shared backbone (6 layers, hidden dimension of 384, and 6 attention heads). **Top (Pretraining):** The backbone is pretrained on nucleotide sequences spanning all eight influenza gene segments (PB2 to NS). **Bottom (Finetuning):** The model is fine-tuned for two distinct tasks: viral evolution prediction (left), which uses an LM head to predict sequences for month $t + 1$ based on historical strains (months $t - 2$ to $t$) from the same region; and subtype classification (right), which employs a classification head to identify the virus subtype based on HA and NA segments.

54,512 genomes (**Figure 1A**). For each virus, the eight segments were simply concatenated in a fixed order (from largest to smallest segment) to form a full-genome sequence without Multiple Sequence Alignment or gaps, and then tokenized (Appendix A.2).

Post-2022 sequences underwent the same quality control procedures and were reserved as retrospective test sets, stratified by subtype and collection month, to enable leakage-free evaluation that mimics real-world forecasting. For geographic generalization, we fine-tuned models on pre-2022 sequences from Europe and Asia and assessed performance on post-2022 genomes from Japan (in-distribution) and the United States (out-of-distribution).

## 4.2 PRETRAINING SETUP

To dissect the contributions of cross-segment dependencies, nucleotide-level variation, genome completeness, and segment-order alignment, we conducted a controlled ablation study using five pretraining variants, each differing in sequence and segment organization; a schematic overview of these configurations is shown in **Figure 2**:

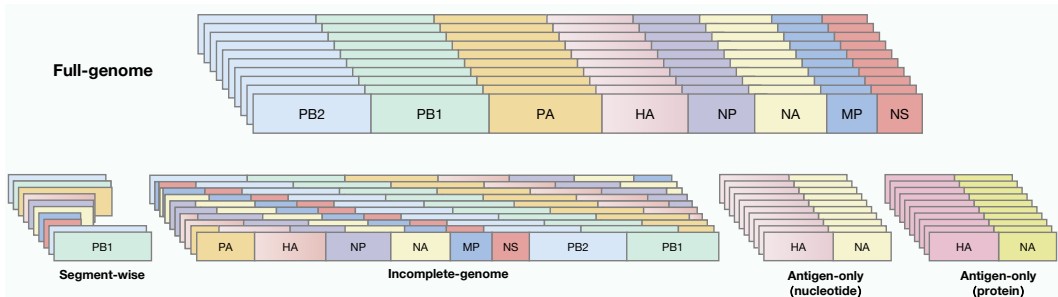

Figure 2: **Pretraining input of AntigenLM and ablation variants.** The standard **Full-genome** strategy (top) concatenates the eight segments from the same isolate in a fixed order. The **Ablation variants** (bottom) explore alternative input formatting: **Segment-wise** uses independent per-segment sequences; **Incomplete-genome** is generated by randomly cropping fixed-length windows from long concatenations, resulting in mixed segments; and **Antigen-only** models restrict input solely to the HA and NA segments using nucleotide and protein (amino acid) sequences, respectively. All the configurations except **Antigen-only (protein)** utilize nucleotide sequences.

1. **Full-genome pretraining:** All eight gene segments (PB2, PB1, PA, HA, NP, NA, M, NS) are concatenated into a single nucleotide sequence (up to ∼13,000 tokens, one nucleotide per token) and trained with an autoregressive objective. This preserves segment ordering and cross-segment dependencies, allowing the model to learn co-evolutionary patterns across the genome.

2. **Incomplete-genome pretraining:** Concatenated genomes are randomly cropped into 12k-token windows while keeping the overall corpus size constant. This disrupts some gene boundaries, allowing us to quantify the effect of losing full functional context.

3. **Segment-wise pretraining:** Each training example contains a single randomly selected gene segment. Corpus size is matched to the full-genome variant, but all cross-segment context is removed, isolating within-segment learning.

4. **Antigen-only (nucleotide) pretraining:** Only the HA and NA segments are included. HA and NA nucleotide sequences are concatenated and trained with the same autoregressive objective. This restricts the model to the two major antigenic determinants while maintaining co-evolution signals between them.

5. **Antigen-only (protein) pretraining:** HA and NA coding sequences are translated into amino acid sequences, concatenated, and tokenized at the residue level. By training directly on HA–NA proteins, this variant focuses solely on antigenic variation in protein space, isolating effects attributable to synonymous nucleotide changes.

All variants use the same backbone, optimizer, learning-rate schedule, number of updates, and effective token budget; random seeds are fixed for data splits, initialization, and shuffling to ensure a controlled comparison. We optimize the standard causal language modeling loss

$$\mathcal{L}_{\text{CLM}} = -\sum_{t=1}^{T-1} \log p(x_{t+1} \mid x_{\leq t}),$$

where $T$ is the sequence length.

## 4.3 FINE-TUNING AND EVALUATION TASKS

We designed four complementary downstream tasks to evaluate AntigenLM's ability to learn biologically meaningful representations and fine-tuned the model accordingly. These tasks assess short- and long-term forecasting, geographic and subtype generalization, and representation quality for classification.

**(1) Next-Month HA/NA Sequence Forecasting.** To evaluate short-term predictive power, AntigenLM was fine-tuned on HA and NA sequences from three consecutive months up to month $t$

and used to predict sequences for month $t+1$ within a given region. Sentinel tokens <HA>and <NA>marked input segments, and <sep>separated sequences from different strains. Outputs were verified for the presence of <HA>and <NA>, and predictions with length deviations exceeding 150 nucleotides were flagged failure. Successful outputs were evaluated for: (i) **Mean AA Mismatch**—average amino acid difference between predicted sequences and the nearest observed strain; (ii) **Epitope-Specific Mismatch**—average mismatch restricted to known epitope sites; and (iii) **Token-Level Perplexity**. Baseline and ablation models were trained and evaluated in parallel.

**(2) Next-Season HA/NA Sequence Forecasting.** For longer-horizon forecasting, AntigenLM predicted sequences for season $T+1$ based on serial sequences from the previous season $T$ (November year $T$ to February year $T+1$). Baselines, including *beth-1*, were trained and evaluated on the same pre-2022 training and post-2022 evaluation splits.

**(3) Geographic and Cross-Subtype Generalization.** To assess out-of-distribution performance, all U.S. sequences were held out during fine-tuning and used exclusively for evaluation. Transferability to the minor subtype H7N9 ($< 5\%$ of pretraining corpus, $< 1\%$ of fine-tuning data) was also tested, demonstrating AntigenLM's ability to generalize to rare subtypes. As no baseline models forecast minor-subtype sequences, comparisons were made against AntigenLM pretraining variants.

**(4) Subtype Classification.** Representation quality was further evaluated by training a lightweight classification head on sentinel token embeddings of HA+NA sequences. Performance was measured using micro-averaged F1 scores.

## 4.4 BASELINE COMPARISON

We benchmarked AntigenLM against three classes of baselines:

**(1) Evolutionary Model — *beth-1*.** *beth-1* is a state-of-the-art site-based dynamic model that estimates mutation fitness and projects site-wise prevalence forward in time. It serves as our primary biological baseline. EVE was excluded as it produces clade-level rather than full-sequence forecasts.

**(2) Tree-Based Predictor — LBI.** The Local Branching Index (LBI) operates on phylogenetic trees derived from HA/NA sequences and has historically informed WHO vaccine strain recommendations, providing a meaningful lower bound for predictive performance.

**(3) General-Purpose DNA/Protein Language Models.** Autoregressive models—HyenaDNA and ProtGPT2—were used to benchmark AntigenLM against general-purpose nucleotide and protein language models. These models are pretrained on large, heterogeneous genomic or proteomic corpora spanning multiple species and genome sizes, allowing us to assess the added value of influenza-specific, biologically informed pretraining.

All language models used identical training/evaluation datasets and matched parameter counts where possible. *beth-1* and LBI were run using publicly available implementations with recommended settings.

## 4.5 IMPLEMENTATION

AntigenLM is implemented in PyTorch using the Hugging Face `transformers` library. Unless otherwise stated, the efficiency statistics in this section refer to the pretraining phase.

We pretrain on 8 NVIDIA A800 GPUs (80 GB each) with a per-device micro-batch size of 1 complete genome and gradient accumulation over 4 micro-steps, yielding an effective global batch size of 32 genomes per optimizer update and a maximum context length of 13,000 tokens. On this setup, the pretraining run processes roughly $7 \times 10^5$ tokens per second across all 8 GPUs, and training the full 54k-genome corpus requires a total compute budget on the order of $10^{18}$ floating point operations, while remaining well within the 80 GB memory budget of each device.

We optimize all models with AdamW (peak learning rate $1 \times 10^{-4}$, linear warmup over the first 5% of updates followed by cosine decay, dropout 0.1, gradient clipping at 1.0). Task-specific fine-tuning

for forecasting and subtype classification reuses the same pretrained backbone and is substantially cheaper than pretraining; we therefore do not report separate efficiency numbers for these stages. Code, preprocessing scripts, and trained checkpoints will be released upon acceptance.

# 5 RESULTS AND ANALYSIS

## 5.1 NEXT-MONTH HA/NA SEQUENCE FORECASTING

We first evaluated AntigenLM on next-month HA and NA sequence forecasting to assess its ability to capture short-term evolutionary dynamics (**Figure 3A**). AntigenLM was fine-tuned on pre-2022 data to predict HA/NA sequences for month t+1 using sequences from the preceding three months (t-2, t-1, t), and evaluated on post-2022 observations. The model produced smooth month-to-month forecasts, with mean amino-acid (AA) mismatches of 3–4 in HA ($< 1\%$ of 566 AAs) and 1–2 in NA ($< 0.5\%$ of 469 AAs). Mismatches within epitope regions were similarly rare and approached zero for NA.

Comparisons with general-purpose language models and ablation controls highlighted the advantages of biologically informed pretraining. (i) HyenaDNA and ProtGPT2 outputs were mostly valid HA/NA sequences with large length deviation and sentinel token missed or misplaced, for which mismatch cannot be calculated; (ii) The incomplete-genome baseline frequently failed to produce valid sequences and exhibited substantially higher AA mismatch rates when it did. (iii) The segment-wise and antigen-only models performed comparably to AntigenLM but showed mild degradation. Together, these results indicate that preserving the integrity of individual segments (at least HA and NA) i s essential for generating valid forecasts, and that incorporating whole-genome, cross-segment context further improves predictive accuracy.

We further quantified model uncertainty using token-level perplexity for all ablation models, except the Antigen-only (protein) variant due to differences in tokenization. AntigenLM achieved a perplexity of 1.26, substantially lower than Incomplete-genome (3.55), Segment-wise (4.42), and Antigen-only (nucleotide) (4.56), indicating superior modeling of the conditional sequence distribution. These results demonstrate that (i) non-HA/NA internal segments contribute meaningful predictive signals (controlled by Antigen-only); (ii) maintaining functional-unit structure improves LM learning (controlled by Segment-wise); (iii) sequence integrity is essential for accurate forecasting (controlled by Incomplete-genome).

## 5.2 FORECASTING NEXT-SEASON CIRCULATING STRAINS

We evaluated AntigenLM on next-season dominant strain forecasting, directly relevant to vaccine strain selection. The model was fine-tuned using pre-2022 sequences from Europe and Asia and tested on post-2022 sequences from Japan, predicting circulating strains for season $T$+1 based on sequences from three consecutive months of season $T$.

**Figure 3B** shows results for H1N1 and H3N2, considering both full-length genes and epitope regions. AntigenLM consistently achieved the lowest amino acid mismatches, with the largest gains observed in H1N1-HA and H3N2-NA, reducing mismatches by over 70% relative to WHO vaccine recommendations (labeled as "current system" in **Figure 3**) and by 50% compared to the site-based model *beth-1*. Site-based models exhibited higher variance and often overpredicted single-site sweeps, reflecting the limitations of their independence assumptions. Improvements were consistent across both full-length and epitope-restricted regions, except for H1N1-NA, which was already below 1, leaving little room for improvement. These results underscore the overall improvement of AntigenLM in forecasting accuracy and highlight its potential utility in guiding vaccine design.

Forecasts were also more seasonally stable, avoiding abrupt clade switches observed in tree-based methods. Across 100 tests (50 per subtype) in Japan, H3N2 strains transitioned to new clades in the next season while H1N1 largely remained within existing clades. AntigenLM accurately predicted sequences of emerging H3N2 clades, consistent with known patterns of antigenic drift. These results demonstrate that AntigenLM provides biologically coherent, actionable predictions, reducing potential mismatches between vaccines and future circulating strains.

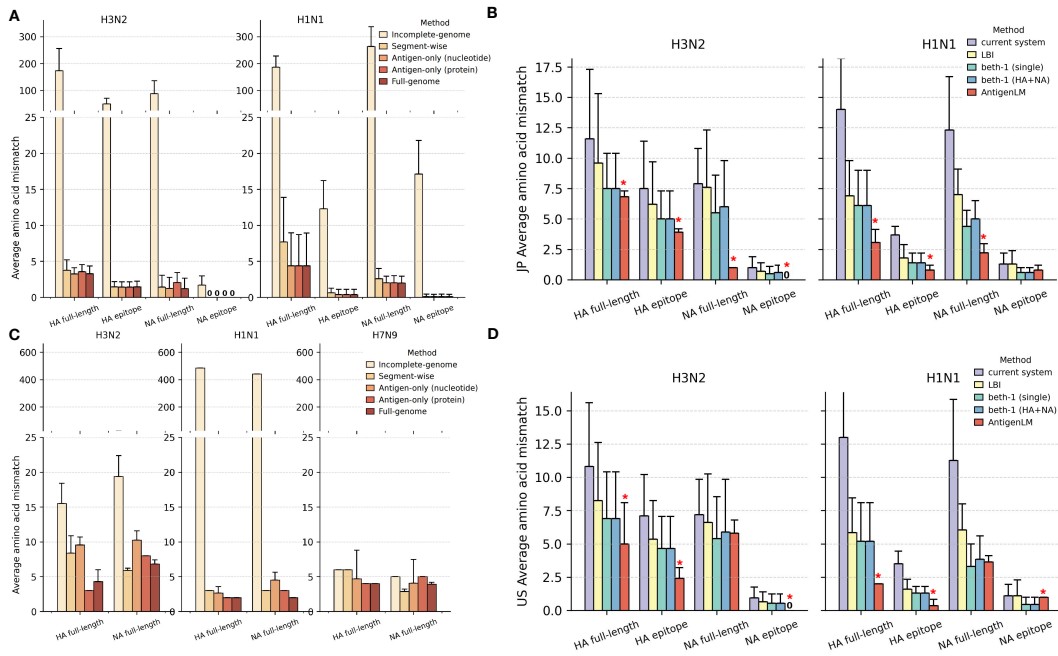

Figure 3: **AntigenLM Achieves the Lowest Amino-Acid Mismatch Across All Forecasting Tasks.** (A) Next-month prediction: AntigenLM (full-genome pretraining) compared with ablation controls. (B) Next-season forecasting on post-2022 Japan data (with pre-2022 data included in fine-tuning): AntigenLM compared with baseline models. (C) Cross-subtype generalization in next-season forecasting: AntigenLM (full-genome pretraining) versus ablation controls for H7N9 prediction. (D) Geographic generalization: AntigenLM evaluated on U.S. data unseen during fine-tuning, compared with baseline models. Asterisks indicate statistical significance (t-test) between AntigenLM and *beth-1*: * $p < 10^{-3}$. Error bars show standard deviations.

## 5.3 GENERALIZATION ACROSS SUBTYPES AND GEOGRAPHIES

Current evolutionary models typically require large amounts of training data and often struggle to forecast emerging strain sequences for minor subtypes or regions with limited historical sampling. Consequently, their utility for predicting the evolution or pandemic potential of emerging strains in under-sampled populations is limited. To assess AntigenLM's ability to generalize beyond its training distribution, we performed two complementary out-of-distribution (OOD) experiments.

**Cross-Subtype Transfer.** We evaluated AntigenLM on the minor influenza A subtype H7N9, which represents only 4.68% of the pretraining corpus and 0.3% (48 sequences) of the fine-tuning set. Despite this extremely limited representation, AntigenLM generated next-season predictions for H7N9 with test counts and mismatch rates comparable to those for the major subtypes H1N1 and H3N2. Interestingly, all ablation models—including the *Incomplete-genome* variant—performed similarly on H7N9 as on the major subtypes, likely due to the higher sequence conservation of H7N9 (Figure 3C). Overall, AntigenLM achieves effective cross-subtype generalization, a capability that remains challenging for current phylogenetic approaches.

**Geographic Generalization.** To evaluate geographic robustness, we held out the U.S. as an unseen region and fine-tuned AntigenLM only on sequences from Europe and Asia. The model maintained substantially lower average AA mismatches than *beth-1* for HA in both H1N1 and H3N2, although improvements for NA were not significant (**Figure 3D**). Analysis of 100 test cases revealed that all involved transitions into clades absent from fine-tuning. AntigenLM correctly predicted clade changes in 90 of these cases (all 50 H1N1 and 40/50 H3N2), though the lack of exposure slightly increased overall mismatch, particularly in NA. Epitope mismatches in NA remained minimal (0–1 amino acid), consistent with in-distribution performance and sufficient for vaccine design. These results indicate that AntigenLM learns global evolutionary constraints rather than memorizing region-specific mutations, enabling reliable predictions in historically under-sampled populations.

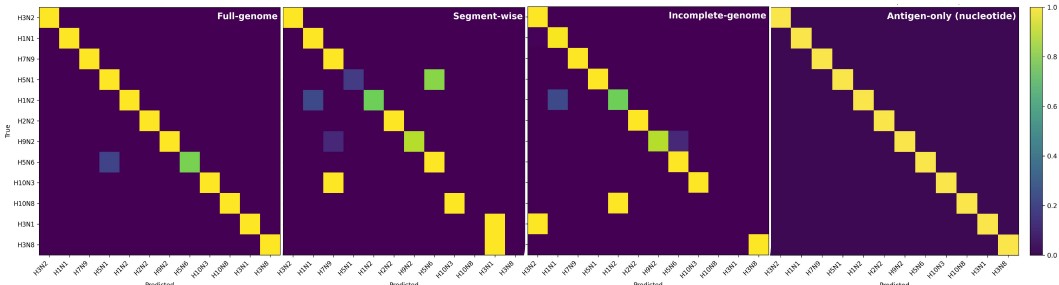

Figure 4: Subtype classification performance of AntigenLM and ablation models. Row-normalized confusion matrices for subtype classification. AntigenLM (left) and Antigen-only (nucleotide) (right) show near-perfect diagonal dominance, indicating highly accurate classification, whereas Incomplete-genome and Segment-wise models (middle) show increased off-diagonal misclassifications, particularly for rare subtypes.

Together, these results demonstrate that AntigenLM does not merely fit dominant subtypes but learns transferable, biologically meaningful representations that generalize across subtypes and geographies—a capability current baselines struggle to achieve. Such robustness is critical for real-world applications where data availability is uneven and novel strains may emerge outside well-sampled populations.

### 5.3.1 SUBTYPE CLASSIFICATION

To further evaluate AntigenLM's multitask capabilities, we tested its performance on subtype classification as a separate downstream task.

Subtype classification is relatively straightforward due to distinct sequence differences among subtypes, and accordingly, AntigenLM and all its variants performed well. The Antigen-only models (nucleotide and protein) achieved 100% accuracy, as expected, since subtypes are solely determined by antigen sequences. Notably, Full-genome AntigenLM also performed exceptionally, achieving a micro-averaged F1 score of 99.81%, with only a single H5N6 strain misclassified as H5N1 among the 530 test strains. All other minor subtypes were classified accurately despite limited training data, demonstrating that AntigenLM's embedding space robustly separates subtypes. In contrast, the Incomplete-genome and Segment-wise variants showed substantially more subtype confusion (**Figure 4**).

These results indicate that AntigenLM learns a well-structured, subtype-aware latent space, supporting applications such as automated influenza surveillance and real-time strain tracking without retraining.

## 6 DISCUSSION AND CONCLUSION

AntigenLM integrates functional-unit preservation into language modeling, substantially improving influenza forecasting. It consistently outperforms general-purpose language models, evolutionary methods, and ablation controls across next-month, next-season, and out-of-distribution tasks, achieving lower amino acid mismatches and biologically coherent predictions that align with subtype transitions. By capturing higher-order dependencies across segments, AntigenLM generalizes across subtypes and geographies, enabling accurate forecasts even in under-sampled populations. Challenges remain for real-time deployment, as forecasts are inherently probabilistic and should complement expert-driven decisions (Ampofo et al., 2011). Overall, AntigenLM demonstrates that integration of biological structure into genomic language models can provide more accurate, interpretable, and actionable insights for viral evolution and public health applications.

## Data and Code Availability

All influenza genome sequences used in this study are publicly available from the GISAID Influenza Virus Resource. The code, data and models are available at AntigenLM .

## Use of Large Language Models

We used large language models only for minor language polishing. No LLMs were used for coding, data processing, experimental design, or analysis, and all reported results were produced by our own code and verified from logged runs.

## Acknowledgment

This work was supported by National Key Research and Development Program of China (2024YFC3405704), the National Science Foundation of China (32371537).

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

# A  DATA USED IN THIS STUDY

## A.1  DATASET

Table 1 summarizes subtype–clade coverage and dataset sizes. Beyond the two dominant subtypes (H3N2 and H1N1), we *intentionally* include many low-resource subtypes—15 of 19 have fewer than 100 sequences—to expose pretraining to broader antigenic/genomic diversity and encourage generalization. These long-tail subtypes are then used mainly to *stress-test transfer* in the subtype-classification finetuning task, where we observe favorable performance despite limited samples. By contrast, our main forecasting experiments do not depend on rare subtypes: they are trained and evaluated on H1N1 and H3N2, with a small extension to H7N9.

In addition, Table 2 provides the temporal distribution of the collected strains, showing coverage from 2012 to 2022. Additionally, Table 3 summarizes the geographical distribution of these pretraining sequences across key global regions such as the USA, China, and Australia, ensuring the breadth and diversity of the data used to train the shared model backbone. Thus, forecasting conclusions are not confounded by extreme class imbalance.

Table 1: Subtype–clade coverage and dataset counts

| Subtype | Clade | Pretrain | Finetune |
|---|---|---|---|
| H1N1 | 6B.1; 6B.1A; 6B.1A.1; 6B.1A.2; 6B.1A.3; 6B.1A.5; 6B.1A.5a; 6B.1A.5a.1; 6B.1A.5a.2; 6B.1A.5a.2a; 6B.1A.5a.2a.1; 6B.1A.5b; 6B.1A.6; 6B.1A.7; 6B.2 | 20680 | 5565 |
| H1N2 | 6B.1; 6B.1A.1; 6B.1A.5a; 6B.1A.5a.2a; 6B.1A.5a.2a.1; 6B.1A.6 | 31 | 74 |
| H2N2 | | 70 | 78 |
| H3N1 | 3C.2a1b.2a.2a.3a.1; 3C.3a | 1 | 4 |
| H3N2 | 3C.2; 3C.2a; 3C.2a1; 3C.2a1b.1; 3C.2a1b.1a; 3C.2a1b.1b; 3C.2a1b.2; 3C.2a1b.2a; 3C.2a1b.2a.1; 3C.2a1b.2a.1a; 3C.2a1b.2a.1a.1; 3C.2a1b.2a.2; 3C.2a1b.2a.2a; 3C.2a1b.2a.2a.1; 3C.2a1b.2a.2a.1a; 3C.2a1b.2a.2a.1b; 3C.2a1b.2a.2a.2; 3C.2a1b.2a.2a.3; 3C.2a1b.2a.2a.3a; 3C.2a1b.2a.2a.3a.1; 3C.2a1b.2a.2a.3b; 3C.2a1b.2a.2b; 3C.2a1b.2a.2c; 3C.2a1b.2a.2d; 3C.2a1b.2b; 3C.2a2; 3C.2a3; 3C.2a4; 3C.3a; 3C.3a1 | 32758 | 8979 |
| H3N8 | 3C.2 | | 3 |
| H5N1 | 1.1.1; 1.1.2; 2.1.2; 2.1.3.2; 2.1.3.2a; 2.1.3.2b; 2.1.3.3; 2.2; 2.2.1; 2.2.1.2; 2.3.2.1a; 2.3.2.1b; 2.3.2.1d; 2.3.2.1e; 2.3.2.1g; 2.3.4; 2.3.4.1; 2.3.4.2; 2.3.4.3; 2.3.4.4b; 3; 7 | 104 | 298 |
| H5N6 | 2.3.4.4; 2.3.4.4b; 2.3.4.4e; 2.3.4.4g; 2.3.4.4h | 25 | 37 |
| H5N8 | 2.3.4.4b | 1 | |
| H6N1 | | 1 | |
| H7N2 | | 1 | |
| H7N3 | | 2 | |
| H7N4 | | 2 | |
| H7N7 | | 2 | |
| H7N9 | | 793 | 1047 |
| H9N2 | | 37 | 78 |
| H10N3 | | | 3 |
| H10N8 | | 4 | 3 |
| **total** | | 54512 | 16169 |

Table 2: Temporal distribution of pretraining strains by subtype

| Subtype | 2022 | 2021 | 2020 | 2019 | 2018 | 2017 | 2016 | 2015 | 2014 | 2013 | Before 2012 |
|---|---|---|---|---|---|---|---|---|---|---|---|
| H10N8 | | | | | | | | | 1 | 3 | |
| H1N1 | 124 | 159 | 2370 | 4918 | 2952 | 1092 | 2122 | 786 | 353 | 295 | 5509 |
| H1N2 | | 2 | 1 | | 4 | 1 | 2 | 1 | | | 20 |
| H2N2 | | | | | | | | | | | 70 |
| H3N1 | | | | | | | | | | 1 | |
| H3N2 | 1472 | 3674 | 963 | 6723 | 4086 | 5470 | 2515 | 2018 | 1117 | 604 | 4116 |
| H5N1 | | | | | | | | 2 | | 7 | 95 |
| H5N6 | | 3 | 2 | | 4 | 3 | 6 | 4 | 3 | | |
| H5N8 | | | 1 | | | | | | | | |
| H6N1 | | | | | | | | | | 1 | |
| H7N2 | | | | | | | 1 | | | | |
| H7N3 | | | | | | | | | | | 2 |
| H7N4 | | | | 2 | | | | | | | |
| H7N7 | | | | | | | | | | 1 | 1 |
| H7N9 | | | | 4 | 1 | 173 | 60 | 68 | 153 | 334 | |
| H9N2 | | | 3 | 11 | 2 | 2 | 6 | 6 | 2 | 2 | 3 |
| **Total** | 1596 | 3838 | 3340 | 11656 | 7051 | 6741 | 4712 | 2885 | 1629 | 1248 | 9816 |

Table 3: Geographical distribution of pretraining strains by subtype

| Subtype | Australia | Brazil | China | France | Russia | Singapore | USA | United Kingdom | others |
|---|---|---|---|---|---|---|---|---|---|
| H10N8 | | | 4 | | | | | | |
| H1N1 | 420 | 375 | 1044 | 638 | 621 | 564 | 9692 | 878 | 6448 |
| H1N2 | | 1 | | 1 | | | 25 | | 4 |
| H2N2 | 1 | | 2 | | 6 | 8 | 27 | 2 | 24 |
| H3N1 | | 1 | | | | | | | |
| H3N2 | 2006 | 1136 | 1283 | 773 | 577 | 403 | 16532 | 1895 | 8153 |
| H5N1 | | | 32 | | | | | | 72 |
| H5N6 | | | 23 | | | | | | 2 |
| H5N8 | | | | | 1 | | | | |
| H6N1 | | | 1 | | | | | | |
| H7N2 | | | | | | | 1 | | |
| H7N3 | | | | | | | | | 2 |
| H7N4 | | | 2 | | | | | | |
| H7N7 | | | | | | | | | 2 |
| H7N9 | | | 791 | | | | | | 2 |
| H9N2 | | | 35 | | | | | | 2 |
| **Total** | 26277 | 14711 | 3217 | 2775 | 2427 | 1513 | 1412 | 1205 | 975 |

A.2    PRETRAINING DATA CURATION AND QC

We retain isolates that provide all eight segments (PB2, PB1, PA, HA, NP, NA, MP, NS).

**CDS parsing and validity (per segment).**

- starts with `ATG` and ends with one of `TAA/TAG/TGA`;
- length is a multiple of 3 and contains no internal stop codons;
- the CDS length falls within an empirically defined sanity range for that segment (outliers and clearly truncated records are removed).

**Additional hygiene.**

- remove records with excessive ambiguous characters;
- ensure subtype annotations are consistent with HA/NA;
- use a fixed concatenation order (PB2→PB1→PA→HA→NP→NA→MP→NS).

### A.3 FINETUNING DATA CURATION (HA/NA-ONLY)

Finetuning uses the same QC criteria as Appendix A.2 *but applied only to HA and NA*. An isolate is **eligible if and only if** its HA and NA CDS pass the validity checks; the other six segments are *not required*, may be missing or incomplete, and are *ignored* even if present.

**Sampling for inputs.** For each region/subtype we select three inputs from the windows in Appendix B (Sep–Oct; Nov–Dec; Jan–Feb). Finetuning models consume only HA/NA tokens (other segments, if any, are unused). The forecasting target is the seasonal consensus defined in Appendix B.

## B FINETUNING SAMPLE SELECTION: THREE-INPUT SETTING

For each target season (October $\rightarrow$ March of year $T+1$), we construct a three-input sample per region/subtype using the *collection date* (ignoring submission delays). Each input must pass the QC in Appendix A.2 and contain at least HA & NA.

Three windows (one sample per window):

- **Sep–Oct**, year $T$. Captures early cross-regional introductions at the onset of the influenza season—the "seeds" that may later establish sustained transmission. Very early months (Jul–Aug) are noisier and less predictive, whereas sampling too late may miss early signals.
- **Nov–Dec**, year $T$. Temperate regions enter winter transmission; lineage-specific growth-rate differences become clearer, and cross-regional flows associated with late-November and December holidays are reflected in the data.
- **Jan–Feb**, year $T+1$. As close as practicable to the WHO Northern Hemisphere vaccine composition meeting, this window captures mid-season replacement events and newly fixed/key substitutions. Information after February is less actionable given manufacturing timelines.

**Target output:** The prediction target for our forecasting experiments aligns with the field's routine answer strain selection: the **nearest sequence**. To meet this target, we construct a three-input sample (balancing lead time and recency). Our model demonstrated **favorable performance** against this standard target, and **superior results** when further tested against the consensus sequence target. This robust finding confirms the validity of our prediction framework.

## C TRAINING DYNAMICS

Under matched optimization and token budgets, Figure 5 reveals consistent differences across the three pretraining variants. **Early burn-in:** the *full-genome* model drops rapidly, *segment-wise* improves more slowly, and *incomplete-genome* decreases minimally. **Mid/late regime:** *full-genome* continues decreasing and converges to the lowest asymptote; *segment-wise* plateaus higher; *incomplete-genome* remains nearly flat, with visually larger bias and limited variance reduction.

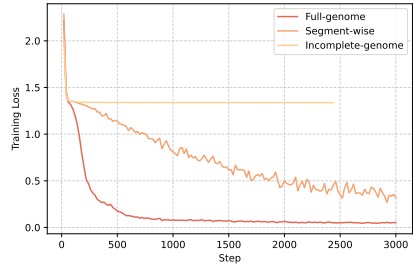

Figure 5: Pretraining loss versus steps for the three variants.

Intact cross-segment context acts as an auxiliary supervisory signal, improving sample efficiency; isolating segments removes these dependencies; cropping random windows mixes unrelated contexts and discards boundaries, weakening the learning signal. All variants share backbone, optimizer, schedule, update count, and token budget; curves are smoothed identically for display and preserve the same ordering without smoothing.

# D  SUBTYPE CLASSIFICATION

We finetune a subtype classifier on top of each pretrained backbone with identical heads and optimization. For qualitative analyses, we extract penultimate-layer embeddings and project them into 2D using the same t-SNE configuration across variants (shared perplexity, initialization, and perplexity-to-sample ratio). t-SNE is used strictly for visualization Figure 6; our main conclusions are based on quantitative forecasting and classification metrics reported in the main paper. For completeness, Appendix Table 4 summarizes clustering metrics (Silhouette, ARI, NMI) computed on the same hidden layer used for the t-SNE plots: the full-genome model achieves the highest Silhouette score and ARI and ties with the segment-wise variant on NMI, indicating more compact and label-consistent subtype clusters than the incomplete-genome and segment-wise baselines. A radar plot of F1 scores across subtypes further emphasizes AntigenLM's superior performance (Figure 7).

Table 4: Clustering metrics for subtype embeddings on the hidden layer used for the t-SNE plots. Higher is better for all metrics.

| Model | Silhouette ↑ | ARI ↑ | NMI ↑ |
|---|---|---|---|
| Full-genome | 0.877 | 0.953 | 0.923 |
| Incomplete-genome | 0.761 | 0.938 | 0.890 |
| Segment-wise | 0.773 | 0.920 | 0.923 |
| Antigen-only (nucleotide) | **0.921** | **0.969** | **0.967** |

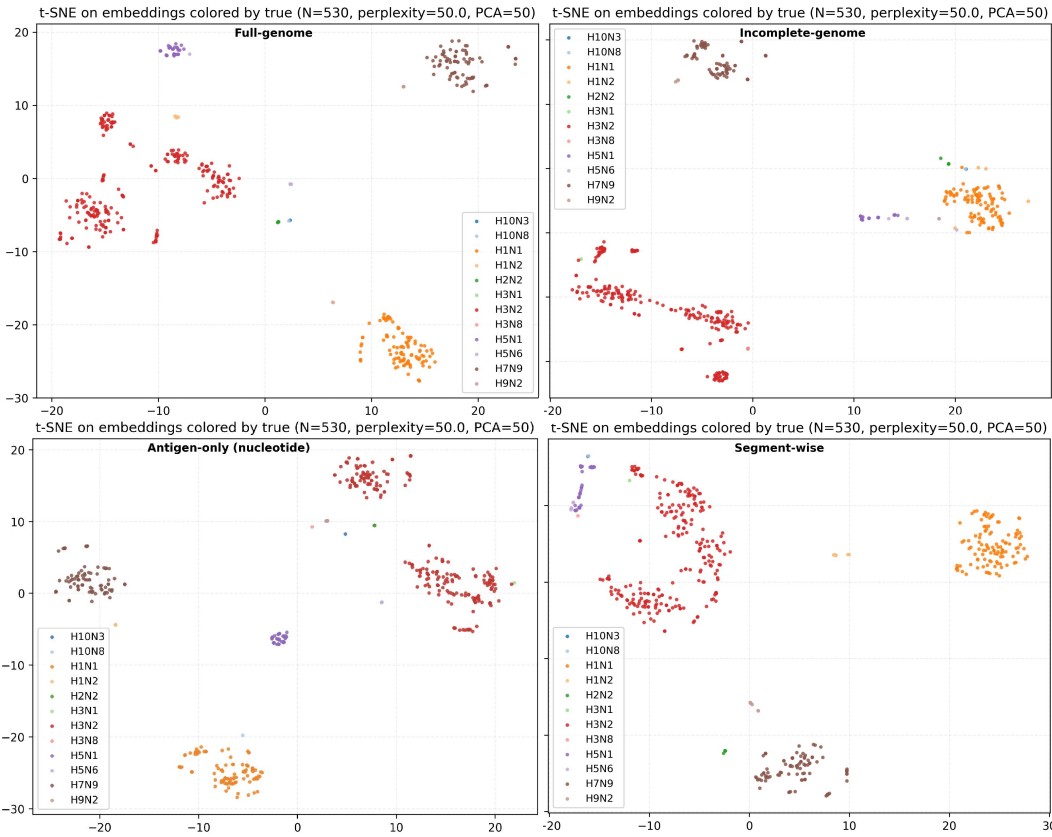

Figure 6: t-SNE of penultimate-layer embeddings from four finetuned subtype classifiers, each initialized from a different pretraining variant. Full-genome and Antigen-only(nucleotide) generally yields the most compact class clusters; Segment-wise is looser; Incomplete-genome lies in between.

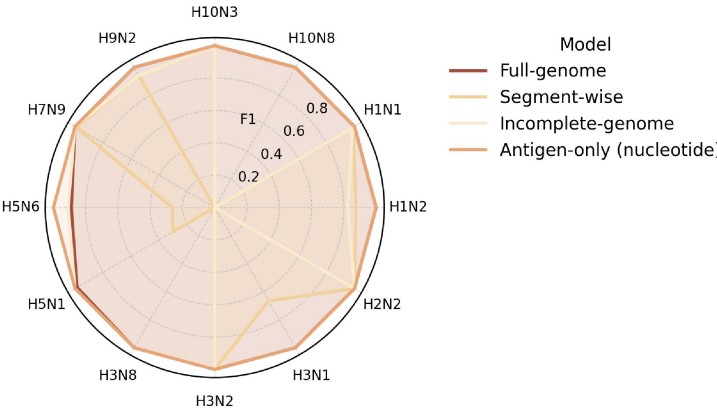

Figure 7: Radar plot of per-subtype F1 scores for AntigenLM (full-genome), Incomplete-genome, Segment-wise models and Antigen-only (nucleotide).

