# OpenReview forum: "AntigenLM: Structure-Aware DNA Language Modeling for Influenza"
_ICLR.cc/2026/Conference — ICLR 2026 Poster_

### Official Review · Reviewer_Vogk · 2025-10-24

**Soundness:** 2
**Presentation:** 2
**Contribution:** 3
**Rating:** 4
**Confidence:** 4

**Summary:**

This paper introduces AntigenLM, a structure-aware DNA language model designed to improve influenza forecasting. It identifies a key limitation: task-specific evolutionary models (like beth-1) often treat mutations as independent events, while general-purpose DNA foundation models are often trained on fragmented, unaligned data, causing them to miss biological context.

To address this, AntigenLM is pretrained on over 54,000 complete influenza A genomes, with each training sample consisting of all eight gene segments concatenated in a fixed order. The hypothesis is that preserving these intact functional units allows the model to learn co-evolutionary dependencies across the entire genome. The model itself is a 6-layer GPT-2-style Transformer with a 13k-token context window to accommodate full genomes.

The evaluation shows that AntigenLM outperforms conventional baselines (beth-1, LBI) in next-season forecasting, reducing amino acid mismatches by over 50% relative to beth-1 on key tasks. Ablation studies show that this Full-genome model outperforms two variants: one trained on Incomplete-genome (randomly cropped) sequences and another on isolated Segment-wise data. The model also shows strong generalization to unseen subtypes (H7N9) and geographies.

**Strengths:**

**(S1)** A clear and important research problem. This work clearly demonstrates the limitations of both site-based evolutionary models and general-purpose genomic foundation models. The hypothesis that a DNA LM for viruses must be pretrained on aligned, intact functional units is intuitive. These arguments thus motivates the need for a new approach.

**(S2)** The integration of biological structure in method. AntigenLM enforces gene- and segment-level boundaries during pre-training and fine-tuning. This offers a compelling biological motivation for functional-unit-aware modeling.

**(S3)** Complete experiment setup. The breadth of the experiments is excellent. It spans several axes such as short-term and long-term sequence forecasting, cross-geographical generalization, rare subtype transfer, and explicit subtype classification. This builds a robust case for the model's capabilities and shows practical utility on real-world scenarios. I particularly respect that some practical considerations like replicate vaccine decision timelines and public health applicability are thoughtfully integrated in experiment framing and data curation.

**(S4)** Interpretability and Visualization. The use of t-SNE visualizations in Fig. 5 and confusion matrices in Fig. 3 offers great interpretability to learned representations. It clearly shows that AntigenLM’s embeddings can yield more distinct and biologically meaningful subtype clusters compared to baselines.

**(S5)** Rigorous Ablation Studies. I respect the design of ablation studies. The comparison of the Full-genome model against its Incomplete-genome and Segment-wise variants counterparts provides a direct evidence of the value of structure preservation. Especially, the performance drop of the ablated models (e.g., perplexity jumping, and near-perfect classification vs subtype confusion) provides a strong support for this paper’s main claim.

**Weaknesses:**

**(W1)** Severe formatting errors (about title). The most immediate and severe issue in my view is that the manuscript is improperly compiled and appears to be in a draft state. The main title of the paper is a placeholder. The actual title, "AntigenLM: Structure-Aware DNA Language Modeling for Influenza," is incorrectly placed as a line of text above the Abstract section. This should be a major flaw for a submission to a top-tier conference. However, given that ICLR permits revisions and iterative author-reviewer discussions, I will reserve final judgment on this shortcoming. I strongly encourage the authors to provide a corrected, complete manuscript in the rebuttal phase.

**(W2)** Incomplete literature review especially for some highly relevant previous work. I encourage the authors to include related discussions in the revised manuscript.

- [1] Learning the Language of Viral Evolution and Escape, Science 2021. This work develops a viral sequence LM directly tied to immune escape and evolutionary prediction for influenza. It is almost exactly the stated target domain here. It helps show what is actually novel in AntigenLM relative to contemporaneous, domain-specialized LMs.
- [2] VQDNA: Unleashing the Power of Vector Quantization for Multi-Species Genomic Sequence Modeling, ICML 2024. This work studies learnable vocabularies which has shown a path beyond fixed tokenizers like BPE and k-mer. Its motivation is similar to AntigenLM which also critiques hand-crafted k-mer and fragmented approaches and offer a learned-representation instead, differing from AntigenLM's explicit functional-unit.
- [3] Dinucleotide Evolutionary Dynamics in Influenza A Virus, Virus Evolution 2019. This work examines sequence-level constraints and evolution, providing insights for the biological basis behind AntigenLM's structural modeling.

**(W3)** Comparisons to DNA foundation model. It is stated that general-purpose DNA foundation models underperform because they are trained on fragmented data. However, none of these models (e.g., ESM, DNABERT, HyenaDNA) are included as baselines for direct comparison. IMHO, the Incomplete-genome model may not be representative enough for these SOTA models, which use different tokenization and pre-training objectives. I encourage the authors to fine-tune at least one genomic LM like DNABERT on their exact downstream tasks and report its performance to support the claim.

**(W4)** In visualization, the interpretive narrative is a bit thin. For example, Fig. 2 showcases reduced mismatch for the AntigenLM variants. But neither the main text nor the caption interprets which features (e.g., epitope conservation vs. overall sequence) drive the model’s superior forecasting. Panel D (geographic generalization) would benefit from a breakdown by outlier scenarios (as mentioned, H3N2-NA had worse US transfer—why?). Similarly, t-SNE visualization in Fig. 5 suggests tighter clusters for the full-genome variant, but quantitative cluster metrics (e.g., inter/intra-class distances) could substantiate claims.

**(W5)** Some unclear method details.

- In Sec. 3.2, while the core causal LM objective is standard, details are missing around the batch sampling strategy, positional encoding modifications for highly variable-length and segmented inputs, and the explicit formulation of the dual-task loss function (especially the weight or scheduling between the two heads).
- Equations in Sec. 4.2 does not clarify if nucleotide padding, masking, or non-coding regions are handled during autoregressive loss computation. These are real concerns in genome application. How are sentinels encoded and differentiated from standard tokens? Are transitions across gene boundaries soft or hard? The method should specify if and how the model penalizes boundary crossing in generation mode.
- The paper states that "All pre-training sequences were aligned by segment". In my view, this could mean (a) the 8 segments for a single virus were simply concatenated in a fixed order, or (b) a full Multiple Sequence Alignment (MSA) was performed across all 54,000 genomes for each segment before concatenation. If (b), this is a massive, non-trivial step, and the method for handling gap tokens is missing. I encourage the authors to clarify this in rebuttal. If it is simple concatenation, please replace the word "aligned" with "concatenated in a fixed order" for precision. If it was an MSA, the full method (tool, gap-token handling) is essential for reproducibility.

**(W6) Minor Points**:

- Model efficiency is described in texts in Sec. 3.3), but no specific metrics such as runtime, FLOPs, or GPU memory footprint.
- The distinction between segment-wise and incomplete-genome variants in Fig. 4 could be more intuitively clarified early on for readers less familiar with genomic ML.

**Questions:**

Most of my major concerns and related recommendations have been stated in the Weaknesses section. I recommend focusing the efforts on addressing those points, as they are critical for strengthening the manuscript in the rebuttal stage.

The following are more specific, minor questions to help the authors think more deeply about certain design choices and experiment setups, which might be helpful for this and future work:

**(Q1)** In Sec. 3.2, how does the model encode sentinel tokens, and how are transitions across gene boundaries handled during both pretraining and decoding? What happens if a generation crosses into a functionally distinct segment? Does this terminate generation, penalize the loss, or trigger any corrective mechanism?

**(Q2)** Beyond t-SNE visualization, are there any more quantitative cluster separation scores or biological validation of embedding space distinctions, especially for rare subtypes?

**(Q3)** Are improvements in Fig. 2 and related tables statistically significant? Is it possible to quantify confidence in reported gains? From example, utilizing p-values or bootstrap intervals.


---
## Justifications:

In summary, this paper presents a promising and intuitive pretraining method for viral genomics. The core idea is strong, and the ablation studies provide clear support for it.

However, the current manuscript seems to be not yet complete and with severe issues. The claims of superiority over other foundation models are not supported by direct comparisons, and the claims about learning co-evolution are unproven. IMHO, these are not minor issues but strike at the overall quality of the paper.

Therefore, I cannot recommend acceptance at this stage. I would be glad to raise my rating if thoughtful responses and improvements are provided in the rebuttal stage. I am also open to follow-up discussions with the authors to help further strengthen this work.

I hope these comments help my fellow reviewers and ACs understand the basis of my recommendation.

---

> ### Author Response · Authors · 2025-11-22
> **Response to Reviewer Vogk**
>
> We thank the reviewer for the thoughtful and detailed feedback, and for recognizing the strengths of **AntigenLM**, including its biological motivation, ablation rigor, and practical utility. We address each concern below.
>
> ## W1 — Formatting / Title
>
> - The manuscript title is placed correctly.
> - All formatting issues in the compiled PDF have been reviewed and corrected.
>
> ## W2 — Literature Review
>
> We sincerely appreciate the reviewer’s suggestions. The recommended references have been incorporated into the revised **Introduction**, where we highlight their relevance to **AntigenLM’s** biological motivation and methodological design.
>
> ## W3 — Comparisons to Foundation Language Models
>
> In response to the reviewer’s request, we added two general-purpose LM baselines (Sec. 4.4):
>
> 1. **HyenaDNA** (DNA LM) fine-tuned on HA/NA forecasting tasks.
>    **Outcome**: Most outputs contained invalid structures, including large length deviation, missing or misplaced sentinel tokens.
>
> 2. **ProtGPT2** (protein LM) fine-tuned on translated HA/NA sequences.
>    **Outcome**: Generated sequences were also largely invalid or biologically implausible.
>
> ### Conclusion:
> General-purpose **LMs** — even when fine-tuned — fail to maintain species-specific, genome-structured constraints that are critical for antigen forecasting. This reinforces the need for biologically grounded, influenza-specific pretraining.
>
> ## W4 — Visualization and Interpretive Narrative
>
> We revised the manuscript to strengthen **interpretability** of downstream results:
>
> - **Epitope vs. full-sequence accuracy** (Sec. 5.2):
>    We clarify that global mismatch improvements extend to **epitope** regions. HA epitopes, especially for H3N2, are more variable than NA, explaining the larger gains seen in HA compared to NA.
>
> - **Geographic transfer** (Sec. 5.3):
>    The higher mismatch in U.S. NA predictions stems from all test viruses transitioning into clades absent during fine-tuning. Despite this, **epitope** mismatches remain extremely low (0–1 aa), which is sufficient for making actionable vaccine recommendations.
>
> ## W5 — Method Details
>
> We clarified several implementation details as requested:
>
> 1. **Batch sampling, positional encoding, and loss functions** (Sec. 3.2):
>    - Sequences are sampled and padded to 13k tokens.
>    - Standard GPT-2 absolute positional encoding is applied.
>    - Pretraining uses an autoregressive LM loss; fine-tuning applies independent task-specific losses.
>
> 2. **Handling padding, masking, non-coding regions, and boundaries** (Sec. 4.2):
>    - Loss is computed only on coding nucleotides.
>    - Padding, non-coding nucleotides, sentinel tokens, and prompt tokens are excluded from the loss.
>    - Sentinel tokens have learned embeddings but are never prediction targets.
>    - Gene boundaries are modeled via sentinel delimiters rather than hard structural constraints.
>
> 3. **Pretraining alignment clarification** (Sec. 4.1):
>    - We replaced “aligned by segment” with the precise description: “segments are concatenated in a fixed order without MSA or gap tokens.”
>
> ## W6 — Minor Points
>
> - **Model efficiency** (Sec. 4.5):
>    We now report computational details: pretraining on 8×A800 80GB GPUs, global batch size 32, 13k context length, ~7×10⁸ tokens/s throughput, and ~10⁸ total FLOPs.
>
> - **Ablation figure**:
>    The ablation figure (formerly Fig. 4) has been moved earlier (now Fig. 2) to improve manuscript flow.
>
> ## Q1–Q3 — Technical Questions
>
> 1. **Sentinel tokens and segment transitions**: addressed in W5.
>
> 2. **Embedding structure**: addressed in W4.
>
> 3. **Statistical significance**:
>    We added Welch two-sample t-tests for per-virus amino-acid mismatch between **AntigenLM** and beth-1 (new Figure. 3).
>
> ## Summary
>
> The revised manuscript now includes:
> - Corrected formatting and title placement
> - Expanded literature review with the suggested references
> - Comparisons against general-purpose LMs
> - Detailed clarifications on batching, masking, and pretraining methodology
> - Improved visualizations and interpretive narratives
> - Statistical rigor through significance testing
>
> We again thank the reviewer for these constructive suggestions, which substantially improve the reproducibility, interpretability, and methodological clarity of the work.

---

> > ### Comment · Reviewer_Vogk · 2025-11-23
> > **Official Response by Reviewer Vogk**
> >
> > To the authors,
> >
> > I would like to thank the authors for their effort put into providing additional results and clarifications during the short rebuttal period. Most of my concerns have been addressed after going through the response and other reviewers' comments.
> >
> > Specifically:
> >
> > - The severe formatting errors and placeholder title have been corrected. The manuscript now looks great.
> >
> > - (my W3 & Other Reviewers) Adding HyenaDNA and ProtGPT2 as baselines is an improvement. The results show that general-purpose LMs struggle to maintain the structural constraints (sentinel tokens, segment ordering) for this specific tasks validates the key motivation for a domain-specific, structure-aware pre-training method.
> >
> > - (W5) Knowing that the model relies on fixed-order concatenation without Multiple Sequence Alignment (MSA) actually strengthens the paper, as it demonstrates the method does not rely on computationally expensive preprocessing for gap handling. The masking and loss on coding regions details are now clear.
> >
> > - (my W4, Q2, Q3) Adding clustering metrics (Silhouette, ARI, NMI) and new statistical significance tests (Welch’s t-test) addresses my concerns about empirical analysis. The results are now more statistically grounded than the original ones.
> >
> > - I noticed the extra "Antigen-only" ablation in the revision. IMHO, this is a valuable addition that empirically supports the biological hypothesis that internal gene segments (non-HA/NA) contribute signal to the evolution of surface proteins.
> >
> > Give these improvements, I raise my rating from (4) to (6). I hope that all these clarifications, promised modifications, and new experiments would be integrated into the final version. This would largely improve the overall quality this work. I also welcome further exchange with the authors (if the rebuttal period permits) and am happy to provide suggestions to help further strengthen this work.
> >
> > Best,
> >
> > Reviewer Vogk

---

> > > ### Author Response · Authors · 2025-11-23
> > > **Response to Reviewer Vogk**
> > >
> > > Dear Reviewer Vogk,
> > >
> > > Thank you for your careful reading, thoughtful feedback, and for revisiting our work during the rebuttal period. We are glad that the clarifications, additional baselines, and new analyses addressed your earlier concerns.
> > >
> > > We greatly appreciate your recognition of these improvements and your acknowledgment of their significance for biological interpretation and methodological rigor—it is very encouraging.
> > >
> > > We are grateful for your increased rating and will ensure that all revisions and new results are fully integrated into the final camera-ready version. We warmly welcome any further discussion or suggestions—within the remaining rebuttal window or beyond—and would be delighted to continue exchanging ideas to further strengthen this work and its broader applications.
> > >
> > > With sincere appreciation,
> > >
> > > The Authors

---

> > > > ### Comment · Reviewer_Vogk · 2025-11-26
> > > > **Follow-up Suggestions by Reviewer Vogk**
> > > >
> > > > To the authors,
> > > >
> > > > Thanks for your reply. I appreciate your responsiveness and your dedication to improving this work. As we approach the final phase of the discussion period, I recommend posting an extra global response once all revisions are complete. IMHO, it would be helpful to list the key points and updates for each reviewer one by one and explicitly note if you think their concerns have been addressed in rebuttal (even if the reviewer has not yet replied). This would also help the ACs understand the current status, your improvements, and evaluate the paper's post-rebuttal quality.
> > > >
> > > > Best,
> > > >
> > > > Reviewer Vogk

---

> > > > > ### Author Response · Authors · 2025-11-26
> > > > > **Response to Reviewer Vogk's Suggestion**
> > > > >
> > > > > Dear Reviewer Vogk,
> > > > >
> > > > > Thank you very much for the thoughtful suggestion and for your engagement throughout the discussion period. We truly appreciate your recommendation to prepare an extra global response summarizing the key updates for each reviewer and explicitly indicating how their concerns have been addressed. This is an excellent idea, and we agree that it will help the ACs gain a clear overview of the post-rebuttal improvements and the current status of the paper.
> > > > >
> > > > > We have followed your advice and prepared detailed global summaries for each reviewer, which we have now posted. Thank you again for the constructive guidance.
> > > > >
> > > > > With sincere appreciation,
> > > > >
> > > > > The Authors

---

### Official Review · Reviewer_HonT · 2025-10-29

**Soundness:** 3
**Presentation:** 2
**Contribution:** 2
**Rating:** 6
**Confidence:** 3

**Summary:**

The influenza virus evolves continuously to evade human adaptive immunity, leading to seasonal epidemics. As a result, influenza vaccine strains must be updated each year to maintain their effectiveness for the upcoming flu season. The paper introduces AntigenLM, an autoregressive DNA language model pretrained on aligned influenza genomes. It robustly captures evolutionary constraints and transfers effectively to the generative tasks of strain forecasting for upcoming influenza seasons. Furthermore, to support the pretraining on aligned genome data, the paper showed that training on unaligned or fragmented gene sequences degrades performance. AntigenLM was compared to an evolutionary model, beth-1, and a tree-based predictor, LBI, on forecasting tasks, where it showed superior forecasting performance on most of the influenza genome segments.

**Strengths:**

- The paper presented a sound way of collecting and preprocessing the influenza genomes, taking care of gene segments alignment and functional unit preservation, which led to an efficient autoregressive model pretraining. Additionally, it has been shown that overlooking alignment and functional unit preservation can lead to performance degradation.
- The proposed model improved influenza forecasting over evolutionary models and tree-based predictors. The model achieved lower genetic mismatches between predicted and observed consensus amino-acid sequences.
- The paper showed geographic and cross-subtype generalization. Specifically, the model was tested on held-out U.S. sequences during finetuning. Additionally, unlike baseline models, the proposed model is able to forecast minor-subtype sequences. The minor subtype sequences were present in pretraining and finetuning, but in much lower amounts (<5% and <1% in pretraining and finetuning corpora, respectively).
- Even though the paper lacks ML novelty, I think the work was technically done well. The data preparation is sound, the model pretraining and finetuning were done effectively, resulting in outperforming the baseline models.
- The authors promised to release code, documentation and data retrieval guidance subject to acceptance.

**Weaknesses:**

- The presentation of the paper requires improvement. The overall writing style and, in particular, the clarity of explanations are of low quality. While the pretraining procedure of the model is well described, the fine-tuning section is poorly explained. The fine-tuning process for each specific task should be described in greater detail. I recommend illustrating the fine-tuning procedures with clear indications of the prompts and the regions being forecasted. The current Figure 1B does not contribute to clarity.
- Although segment-wise pretraining is a reasonable approach, the incomplete-genome pretraining with random cropping as a baseline model appears conceptually weak. It would be more meaningful to pretrain on incomplete genomes while excluding entire segments that are not used during fine-tuning and forecasting, retaining only the complete HA and NA segments in their correct order. This would demonstrate whether information from other segments contributes meaningfully, or if comparable performance can be achieved using only the properly aligned HA/NA segments.
- In the segment-wise pretraining, all segments—including those not used during fine-tuning and inference—are utilized. The non-HA/NA segments are likely less informative. The authors should elaborate on their choice of baseline models (segment-wise pretraining and incomplete-genome pretraining) and clarify why these represent sound alternative approaches. At present, the baseline models appear to have been selected post hoc, possibly after the development of AntigenLM, merely to provide comparison points.
- Epitopes are specific regions within influenza HA/NA segments that are targeted by the host immune system. While AntigenLM outperforms beth-1 and LBI on HA epitopes, the authors do not discuss the superior performance of beth-1 on NA epitopes. This omission should be addressed.

**Minor comments:**
- There are typos in Fig. 1B. "Finetuneing" should be written as "finetuning."
- In Fig. 1B, the heads (classification and LM) blocks and the captions do not align with the text lines 134 -- 138.

**Questions:**

1. Please comment and elaborate on the choice of the baseline pretraining methods. It's important to include this in the paper.
2. Please comment on the superior performance of beth-1 on NA epitopes. It's important to include this in the paper as well.
3. How exactly was fine-tuning for next month and next season done? During inference, what was a prompt? Could you elaborate on this in the paper and illustrate, if possible?

---

> ### Author Response · Authors · 2025-11-22
> **Response to Reviewer HonT**
>
> We thank the reviewer for the constructive and encouraging feedback. Below we address each point concisely.
>
> ## W1 — Presentation and Clarity
>
> We appreciate the suggestion and substantially improved fine-tuning descriptions:
>
> - **Sec. 3.4 (new)** now clearly explains:
>   1. how historical HA/NA blocks condition forecasting,
>   2. how sentinel tokens operate during fine-tuning and inference,
>   3. how subtype logits are produced for classification.
>
> - **Sec. 4.3** now explicitly defines the next-month and next-season forecasting tasks, including temporal windows and evaluation setup.
> - **Fig. 1B** has been updated to illustrate the full fine-tuning workflow, including sentinel-token placement and forecasted regions.
>
> ## W2 + W3 — Baseline Rationale and Role of Non-HA/NA Segments
>
> We agree that the contribution of non-surface segments should be evaluated directly. The revision now includes **Antigen-only ablations** (Sec. 4.2):
>
> ### Models:
> - **Antigen-only (nucleotide)** and **Antigen-only (protein)** pretrained solely on concatenated HA+NA.
> - Fine-tuned and evaluated identically to **AntigenLM**.
>
> ### Findings:
> - **Antigen-only models** perform slightly worse than the **Full-genome model** but outperform **Incomplete-genome** and **Segment-wise** ablations.
> - This demonstrates:
>   1. non-HA/NA internal segments contribute meaningful predictive signals (controlled by **Full-genome**);
>   2. maintaining functional-unit (segment) structure improves LM learning (controlled by **Segment-wise**);
>   3. sequence integrity is essential for accurate forecasting (controlled by **Incomplete-genome**).
>
> ## W4 — NA Epitope Performance
>
> We appreciate the emphasis on **epitope-level mismatches**:
>
> - **H1N1-NA**: Highly conserved; all models achieve ~1 aa mismatch, leaving limited headroom. **AntigenLM** is comparable to beth-1.
> - **H3N2-NA**: **AntigenLM** achieves 0 mismatches, outperforming beth-1 and demonstrating accurate modeling of NA-specific constraints.
>
> We added this discussion in **Sec. 5.2**, emphasizing its relevance for vaccine-strain selection.
>
> ## Minor Fixes
>
> - Corrected typos (e.g., **Finetuneing** → **finetuning**).
> - Updated **Fig. 1B** for consistency (head block alignment, captions).
>
> ## Q1–Q3
>
> These are fully addressed by the updates for **W2+W3** (baseline rationale), **W4** (epitope performance), and **W1** (fine-tuning clarity).
>
> ## Closing
>
> We again thank the reviewer for the thoughtful feedback. The revised manuscript now includes:
> - Clearer fine-tuning descriptions (**new Sec. 3.4**)
> - Explicit baseline rationale and **Antigen-only ablations** (**new Sec. 4.2**)
> - Detailed **NA epitope analysis** (**Sec. 5.2**)
> - Improved figures and overall clarity
>
> These revisions strengthen the biological grounding, methodological rigor, and interpretability of **AntigenLM**.

---

> > ### Comment · Reviewer_HonT · 2025-11-23
> > **Response to authors and additional questions**
> >
> > Thank you for addressing my comments and questions. I think the presentation is now clearer, and the paper is strengthened.
> >
> > **Additional comments:**
> > - Based on Figure 3A, it seems the antigen-only models perform comparably to the full-genome model on the next-month prediction problem. However, in Figure 3C, it is not the case. Could you please add to the paper the criteria for selecting the final model configuration? Was it based on token-level perplexity?
> > - Please add the antigen-only models as ablation models to the subtype classification task (Figure 4 and Figure 6).
> >
> > I believe the authors did a good job in addressing my and other reviewers' comments, and the paper is more suited to be presented at the conference. If my remaining comments are clarified and addressed, I will raise my score!

---

> ### Author Response · Authors · 2025-11-23
> **Response to Reviewer HonT’s Additional Questions**
>
> Dear Reviewer  HonT,
>
> Thank you for your thoughtful feedback and for reviewing our revisions. We are glad that the clarifications have made the presentation clearer and strengthened the manuscript.
>
> Regarding your additional comments:
>
> ## 1. Inconsistent performance of Antigen-only model and configuration selection
>
> Thank you for highlighting this point.
>
> ### Why Antigen-only performs differently in Fig. 3A vs. Fig. 3C
>
> Figure 3A evaluates **next-month, within-subtype** prediction — an easier setting where HA/NA evolution is highly autoregressive. In this regime, antigen-only models capture most short-term signals and therefore appear close to the full-genome model.
>
> Figure 3C examines **next-season and cross-subtype** forecasting, which is substantially more challenging and depends on long-range **genomic** constraints. In these settings, information from the rest of the genome — such as inter-segment co-adaptation, packaging signals, and genome-integrity constraints — becomes essential. As a result, the **full-genome model consistently outperforms** the antigen-only and other ablation variants.
>
> ---
>
> ### Model-selection protocol
>
> Our final **AntigenLM** configuration was selected based on **overall downstream forecasting performance across all tasks**, rather than token-level perplexity alone. All ablations were trained and evaluated on identical data splits. The full-genome model was chosen because it achieved the strongest combined performance across next-month, next-season, and **cross-subtype** forecasting, full-antigen and **epitope-mismatch** metrics, and token-level perplexity.
>
> ---
>
> ## 2. Evaluation of the Antigen-only model in subtype classification
>
> We did evaluate **subtype** classification for the antigen-only model. It achieved **100% F1**, and the results are now included in Fig. 4 and Fig. 6. This high accuracy is expected: subtype classification is intrinsically straightforward, as HA/NA **subtypes** differ by 20–40% at the nucleotide level, and correct segment assembly leads real-world performance to approach 100%.
>
> ---
>
> We hope these clarifications address your remaining points, and we welcome any further suggestions to strengthen the work.
>
> With appreciation,
>
> The Authors

---

> > ### Comment · Reviewer_HonT · 2025-11-24
> >
> > Thank you for addressing my remaining points.
> >
> > I think the paper is now a valuable contribution to the community. Thus, I'm raising my rating.

---

> > > ### Author Response · Authors · 2025-11-25
> > > **Response to Reviewer HonT**
> > >
> > > We truly appreciate your thoughtful reassessment of our revision and are grateful for the improved rating. Your encouraging feedback means a lot to us and inspires us to further refine our work.

---

### Official Review · Reviewer_be7j · 2025-10-30

**Soundness:** 2
**Presentation:** 1
**Contribution:** 1
**Rating:** 2
**Confidence:** 4

**Summary:**

This paper presents AntigenLM, a transformer-based DNA language model designed to predict how influenza viruses evolve. Instead of treating genomes as random strings of nucleotides, the model keeps the natural structure of the virus—eight gene segments joined in the right order—so it can learn both local and global relationships. The authors pretrain AntigenLM on more than 50,000 influenza A genomes and then fine-tune it to forecast future HA and NA sequences, which are key for vaccine design. They compare it with existing evolutionary and sequence models like beth-1 and show that AntigenLM gives lower amino-acid mismatches and better subtype classification. Ablation studies also suggest that keeping the full genome intact during pretraining matters for accuracy. Overall, it’s a clear attempt to bring structure awareness into DNA foundation models for real biological forecasting.

**Strengths:**

The idea of preserving full functional units during pretraining feels biologically meaningful and easy to justify. It makes sense that genomes should be learned as whole systems, not chopped-up pieces. The model is compact and computationally reasonable, which makes the method more accessible to other labs. The overall contribution to treating DNA as structured language is simple but potentially important for future genomic foundation models.

**Weaknesses:**

1. The paper claims a big jump over existing models, but I’m not fully convinced it’s a fair comparison. The baselines feel a bit old — why not include newer genomic LMs like HyenaDNA or TITAN? That would make the improvement more believable.

2. Figure 1 is visually appealing and central to understanding the paper, but it could better communicate data imbalance, temporal coverage, and architectural specifics. A supplementary figure showing training sample counts per region/year and a more explicit schematic of sentinel-token flow would improve interpretability.

3. The functional-unit aware idea sounds interesting, but it feels mostly like a data-organization trick, not a new modeling method. Could similar results be achieved just by using longer context windows?

4. The results in Figure 2 look visually impressive, but some doubts arise. The reported 70% mismatch reduction seems unusually large, given that beth-1 already performs well on H1N1/H3N2; could these differences stem from dataset splits or evaluation metrics rather than genuine model ability? Error bars are wide and overlap in several cases, yet statistical significance isn’t shown. It’s also unclear whether test sets are truly unseen in time (to avoid leakage) or whether the consensus targets overly smooth the diversity of real viral populations.

5. H1N1 and H3N2 dominate the dataset, while minor subtypes like H7N9 have extremely small test sets. The paper interprets success on H7N9 as cross-subtype generalization, but with <50 examples, this could just reflect memorization or noise.

6. The paper doesn't seem to have been revised well. The title of the paper is not written in the correct place. Figure 2 caption includes the typographical symbol “¿70%” instead of a proper “~70%” or “≈70%.” Appears to be a stray inverted question mark (encoding artifact) at p.7, caption line. Add proper appendix separation and consistent figure numbering.

7. The model architecture (6 layers, 384 dim) is very small compared to typical DNA LMs. Are we sure it isn’t underpowered — or that the gain isn’t just from better preprocessing?

8. The authors say the model “scales to population-level datasets,” but training was done on ~54 k genomes. That’s not really population scale; what happens at millions?

9. The “near-perfect” subtype classification (99.8 F1) seems almost too good to be true. Was the classifier evaluated on strictly unseen subtypes or just held-out samples of known ones?

**Questions:**

1. The t-SNE plots in the appendix look clean, but t-SNE always makes clusters look nice. Could we see a quantitative metric instead of a pretty visualization?

2. There’s no comparison to protein-level models (e.g., ESM-2). Would those perform equally well or even better for HA/NA sequence forecasting?

3. In the results, H3N2-NA seems slightly worse under U.S. testing. Any insight why that happens? Could it reveal overfitting to regional sequence biases?

---

> ### Author Response · Authors · 2025-11-22
> **Response to Reviewer be7j part 1of 2**
>
> We sincerely thank the reviewer for the thoughtful and constructive feedback. We understand the initial skepticism regarding the magnitude of improvement; we shared this reaction before conducting extensive ablations and controls. Below, we summarize the biological and methodological foundations underlying the results and address each point.
>
> ## Overall Clarification
>
> The gains over the “current system” for WHO vaccine-strain selection and the prior SOTA (beth-1) arise from biological constraints that site-wise models fail to model:
>
> 1. **Long-range genomic constraints:** Inter-segment RNA–RNA interactions, reassortment patterns, and co-adaptation shape HA/NA evolution but are not captured by site-wise models or by general-purpose LMs without explicit genome-structure learning.
> 2. **Genome-wide nucleotide signals:** Synonymous changes and non-coding sequences provide predictive signals missing from protein-only models.
> 3. **High-quality complete genomes:** Our stringent full-genome quality filters reduce errors and noise, enabling learning of clean evolutionary structure—typically infeasible for general-purpose LMs.
>
> To clarify these points, we added general-purpose LM baselines and Antigen-only ablations (protein and nucleotide) and expanded methodological details as suggested.
>
> ## W1 — Baselines with State-of-the-Art Language Models
>
> We included two strong general-purpose autoregressive LMs (Sec. 4.4):
>
> - **HyenaDNA (DNA LM):** Fine-tuned on HA/NA; outputs were largely invalid (missing/misplaced sentinel tokens, severe length drift).
> - **ProtGPT2 (protein LM):** Fine-tuned on translated HA/NA; outputs also degraded with sentinel corruption.
>
> Both fail to maintain species-specific, whole-genome evolutionary constraints, underscoring the importance of **AntigenLM’s** functional-unit-aware, full-genome nucleotide pretraining.
>
> ## W2 — Figure 1 Improvements
>
> We revised Figure 1 and related materials:
>
> - Revised Appendix A.1 showing sample counts explicitly across regions and years.
> - Updated Fig. 1B to clarify sentinel-token placement and their usage during fine-tuning and inference under a standard GPT-2 framework.
>
> ## W3 — Functional-Unit-Aware Modeling vs. Longer Context
>
> Segment-aware modeling is not merely organizational; it encodes biological structure. Ablations show:
>
> - Randomly chopping genomes to similar lengths (~12 kb) markedly harms performance.
> - Very long-context LMs (e.g., HyenaDNA with >1M-token windows) still fail to reconstruct segment-level constraints.
>
> Thus, improvements arise from functional-unit structure, not context length alone.
>
> ## W4 — Performance and Evaluation Rigor
>
> We clarified evaluation practices:
> - The ~70\% mismatch reduction is relative to the **current system** (WHO adopted), not beth-1.
> - Splits are strictly temporal; no leakage is possible.
> - Welch’s t-tests between **AntigenLM** and beth-1 were added (Fig. 2 / revised Fig. 3).
> - We report mismatch to (1) nearest individual test strains (beth-1 protocol) and (2) cluster-level consensus sequences; **AntigenLM** improves over all baselines in both settings.
>
> ## W5 — Reliability for the Minor Subtype (H7N9)
>
> We used 82 H7N9 test sets (4 strains each), comparable to the 100 tests for H1N1/H3N2. H7N9 evolution is relatively conserved, explaining stronger ablations performance. The key point is that **AntigenLM** generalizes well despite limited historical data—an area where current systems and site-wise models struggle. Larger curated datasets will further improve robustness.
>
> ## W6 — Formatting and Figure Issues
>
> All noted issues were corrected:
>
> - Title placement.
> - “~70\%” → “~70\%”
> - Appendix restructuring and consistent numbering.
>
> ## W7 — Model Capacity
>
> **AntigenLM** is purposefully lightweight (6 layers, 384-dim). Influenza A’s compact (~13 kb) genome and strong internal structure make a small, domain-specific LM appropriate. Ablations confirm that benefits come from functional-unit-aware pretraining and temporal fine-tuning—not from model size.
>
> ## W8 — Scalability to Population-Level Datasets
>
> “Population-level” here refers to tens of thousands of complete genomes, typical across many species. **AntigenLM** scales efficiently due to:
>
> - Compact genome size.
> - Lightweight architecture.
> - Segment-aware batching.
> - Autoregressive training that naturally extends to millions of genomes.
>
> Performance is expected to improve as high-quality datasets expand.
>
> ## W9 — Near-Perfect Subtype Classification
>
> Subtype classification is intrinsically straightforward:
> - HA/NA subtypes differ by 20–40\% at the nucleotide level.
> - Partial HA/NA fragments (>300 bp) typically suffice.
> - When segments are assembled correctly, real-world accuracy approaches 100\%.
>
> Controls confirm this:
> - **Antigen-only ablations** reach 100\% F1.
> - Fine-tuned **HyenaDNA** achieves 99.81\%, matching **AntigenLM**.
>
> Thus, **AntigenLM’s** 99.81\% F1 is entirely expected and not due to leakage or overfitting.

---

> ### Author Response · Authors · 2025-11-22
> **Response to Reviewer be7j part 2of 2**
>
> ## Q1 — t-SNE Artifacts
>
> To validate t-SNE, we report quantitative clustering metrics:
>
> - Silhouette: 0.88 (full) vs. 0.76–0.77 (ablations)
> - ARI / NMI: 0.95 / 0.92
>
> These confirm that the full model produces genuinely tighter, label-consistent clusters.
>
> ## Q2 — Protein-Level Model Comparison
>
> Addressed in W1.
>
> ## Q3 — Regional Variations in H3N2-NA (U.S. Transfer)
>
> All 100 U.S. transfer cases transitioned into clades absent from fine-tuning data. Despite this, **AntigenLM** predicts the correct clade shift in 90/100 cases.
>
> **AntigenLM** > beth-1 on HA and ties on NA.
>
> - NA epitope mismatch remains 0–1 aa, which is sufficient for vaccine design.
>
> These results show that **AntigenLM** retains biologically meaningful constraints even under clade shift.

---

> ### Author Response · Authors · 2025-11-26
> **Follow-up on Rebuttal**
>
> Dear Reviewer be7j,
>
> We are writing to kindly follow up on our responses to your insightful comments and feedback. In our rebuttal, we have provided detailed clarifications and additional analyses addressing your concerns and suggestions.
>
> If there are any remaining questions or points you would like us to address further, please do not hesitate to let us know. Your feedback has been invaluable in refining our work, and we would greatly appreciate your thoughts on whether our responses meet your expectations.
>
> Thank you once again for your time and effort in reviewing our submission.
>
> Sincerely,
>
> The Authors

---

> > ### Author Response · Authors · 2025-11-27
> > **Follow-up on Rebuttal (Request for Feedback)**
> >
> > Dear Reviewer be7j,
> >
> > We hope this message finds you well. We would like to kindly follow up on our earlier rebuttal to your detailed and insightful comments. In our responses, we provided additional analyses, new baselines, expanded methodological clarification, and revised figures addressing each of your concerns.
> >
> > If there are any remaining questions or points you feel require further clarification, we would be grateful to address them. Your feedback has been instrumental in improving the manuscript, and we would very much appreciate hearing whether our revisions satisfactorily resolve the issues you raised.
> >
> > Thank you again for the time and care you have devoted to reviewing our work.
> >
> > Sincerely,
> >
> > The Authors

---

### Official Review · Reviewer_DjYH · 2025-10-30

**Soundness:** 2
**Presentation:** 3
**Contribution:** 3
**Rating:** 6
**Confidence:** 3

**Summary:**

The authors developed a new DNA language model, AntigenLM, for predicting the evolution of influenza genomes. This is the first work on applying DNA LM to this task to my knowledge. The training data appear carefully constructed, and the evaluation tasks were well designed. It showed notable performance gains over traditional methods. Although I have some concerns regarding excluding protein language models from the benchmark, I overall recommend a weak acceptance of this work.

**Strengths:**

1. It is a novel idea to use a DNA language model for predicting viral evolution, which is an important problem for public health.
2. There are very few DNA language models for eukaryotic viruses, and this work has filled this gap.
3. The training data is carefully curated and documented in detail. It could be a useful resource to the community.
4. The evaluation tasks are well-designed, and data splitting was done thoughtfully.

**Weaknesses:**

1. I am not sure if the authors covered all the appropriate baselines. I am not super familiar with the field, but there seem to be many works on using protein language models to predict viral evolution. For example, https://www.science.org/doi/10.1126/science.abd7331, https://www.nature.com/articles/s41392-024-02066-x, https://www.biorxiv.org/content/10.1101/2025.08.04.668423v1. Although this model is trained at the DNA level, the evaluations seem mostly at the amino acid level. The authors should justify excluding those models from the comparisons.
2. Following up on the previous point, the authors should also demonstrate the necessity or advantage of modeling at the DNA level rather than the protein level.
3. The description of the model architecture is obscure. For example, does it have 6 heads per layer or 1 head per layer? A more detailed diagram of the architecture might be helpful.

**Questions:**

1. Is the sentinel token not used during pretraining?
2. I am a bit confused about how the notion of time is built into the model. For example, in the next-month prediction, is the model just fine-tuned on sequences at month t, and then used to generate sequences supposedly for month t+1 without any constraint or guidance?

---

> ### Author Response · Authors · 2025-11-22
> **Response to Reviewer DjyH**
>
> We thank the reviewer for the constructive and insightful comments. Below we address each point in turn.
>
> ## W1 + W2 — Necessity of DNA-level modeling and inclusion of protein LMs as baselines
>
> Thank you for raising this important question. In the revision, we:
> - added protein language model (PLM) baselines.
> - clarified why DNA-level modeling is essential for influenza forecasting.
>
> ### Added PLM baselines (Sec. 4.2, 4.4):
> - **ProtGPT2** — autoregressive protein generator, fine-tuned on translated HA/NA proteins matching AntigenLM's training split.
> - **Antigen-only (protein) pretraining** — AntigenLM variant pretrained on HA/NA amino-acid sequences and fine-tuned with the same forecasting objective and data split.
>
> ### Results (Sec. 5.1, Fig. 3):
> - ProtGPT2 generated mostly invalid antigens (e.g., lost or misplaced sentinel tokens, large length deviations).
> - Protein-only AntigenLM produced syntactically valid proteins but consistently showed higher AA mismatch than AntigenLM.
>
> These findings align with biological understanding: influenza evolution is strongly governed by nucleotide-level constraints and genome-wide interactions, such as synonymous and noncoding mutations affecting RNA structure, packaging signals, codon adaptation, and co-adaptation with polymerase PB1-PB2. Protein-only models cannot access these signals and treat HA/NA in isolation, preventing them from maintaining structural and positional constraints required for realistic antigen evolution. Additionally, protein-level pretraining requires CDS extraction and translation, imposing assumptions and a heavier preprocessing burden that DNA-level modeling avoids.
>
> ### Differences from other protein LMs:
> - **Science 2021** and **Nature Signal Transduction 2024** focus on antibody escape/fitness prediction, requiring structural/experimental inputs that are not available for future strains and therefore unsuitable for forecasting.
> - **PLANT (biorxiv 2025)** learns HA embeddings only; it is not generative and cannot produce full HA/NA sequences.
>
> We have updated the manuscript to include these baselines and clarified the rationale for DNA-level modeling in the Introduction.
>
> ## W3 — Model architecture clarification
>
> We appreciate the request for detail. Each Transformer layer in AntigenLM contains **6 attention heads**, with 384 hidden dimensions and a 1,536-dimensional FFN. Figure 1B has been updated, and Sec. 3.1 now provides a clearer architectural description.
>
> ## Q1 — Use of the sentinel token
>
> **Clarification added to Sec. 3.2:**
> The sentinel token appears only during fine-tuning (for mask-span prediction).
>
> ## Q2 — Incorporation of temporal information
>
> We apologize for the earlier ambiguity. The model does not generate future sequences without conditioning. Temporal structure is provided explicitly:
> - **Next-month prediction:** concatenate HA/NA sequences from the three months prior to time `t` in chronological order; fine-tuning then teaches the model to predict month `t+1`, all strains collected from the same geographic region.
> - **Next-season prediction:** same construction, but outputs correspond to season `T+1` rather than month `t+1`, according to strains collected in months of season `T`.
>
> Sec. 4.3 now clearly describes this time-conditioned forecasting setup.
>
> We thank the reviewer again for the thoughtful feedback. Your comments significantly improved the clarity of the manuscript, particularly regarding protein baselines, architectural transparency, and temporal conditioning. We believe the revised version fully addresses the concerns raised.

---

> ### Author Response · Authors · 2025-11-26
> **Follow-up on Rebuttal**
>
> Dear Reviewer DjyH,
>
> We are writing to kindly follow up on our responses to your insightful comments and feedback. In our rebuttal, we have provided detailed clarifications and additional analyses addressing your concerns and suggestions.
>
> If there are any remaining questions or points you would like us to address further, please do not hesitate to let us know. Your feedback has been invaluable in refining our work, and we would greatly appreciate your thoughts on whether our responses meet your expectations.
>
> Thank you once again for your time and effort in reviewing our submission.
>
> Sincerely,
>
> The Authors

---

> > ### Comment · Reviewer_DjYH · 2025-11-26
> >
> > Thank you for addressing the comments from me and other reviewers. I think the added experiments and the updated manuscript have substantially increased the clarity and soundness of the study. My requests above are mostly addressed in the authors' response. Hence I am raising my score.

---

> > > ### Author Response · Authors · 2025-11-27
> > >
> > > Thank you for your thoughtful follow-up and for re-evaluating our work. We appreciate your recognition of the added experiments and clarifications, and we are grateful that our revisions addressed your concerns. Thank you as well for raising your score—your feedback has been very helpful in strengthening the paper.

---

### Author Response · Authors · 2025-11-22
**Overall Response and Revision Summary**

We sincerely thank all reviewers for their constructive feedback. The reviewers recognized that **AntigenLM** represents a major advance by integrating biologically informed, full-genome language modeling with practical applications in influenza vaccine design. Its key contributions include:

1. Accurate forecasting of HA/NA sequences across **subtypes**, including minor or emerging strains.
2. Near-zero **epitope** mismatches, critical for vaccine strain selection.
3. Modeling long-range inter-segment and nucleotide-level dependencies that site-wise or general-purpose LMs cannot capture.

These features position **AntigenLM** as both a **methodological advance** in genome-scale language modeling and a **practical tool** for vaccine design.

Based on reviewer comments, we extensively revised the manuscript as summarized below:

## 1. Baseline Rationale and Ablations
- Introduced biological mechanisms of whole-genome nucleotide-level constraints on HA/NA evolution to clarify AntigenLM’s rationale.
- Added comparisons to general-purpose LMs (HyeaDNA, ProtGPT2) showing that performance gains arise from functional-unit-aware, full-genome pretraining.
- Expanded Antigen-only ablations (nucleotide and protein) to quantify contributions from non-HA/NA segments and genome integrity.

## 2. Methodology and Fine-Tuning Clarity
- Added Sec. 3.4 describing forecasting and classification fine-tuning; conditioning on historical HA/NA, sentinel-token usage, and classification head outputs.
- Updated Sec. 4.3 to explicitly define next-month and next-season tasks with temporal windows.
- Updated Fig. 1B to illustrate sentinel-token placement and forecast regions.
- Clarified study design, data splits, and evaluation procedures.

## 3. Evaluation and Results
- Detailed NA **epitope-level** performance, showing near-zero mismatches for H3N2 and consistent robustness across **subtypes**, including H7N9.
- Added statistical tests and **quantitative embedding metrics** to strengthen evaluation rigor.

## 4. Figures and Presentation
- Revised Fig. 1B for clarity.
- Clarified model implementation and capacity.
- Supplemented data distribution across regions and years.
- Corrected minor typos and formatting issues.

These revisions **clarify the mechanisms underlying AntigenLM’s improvements** and strengthen its reliability, **interpretability**, and applicability for both genome-scale language modeling and practical vaccine design.

---

### Author Response · Authors · 2025-11-26
**Extra Global Response (Post-Rebuttal Summary of Revisions)**

We thank all reviewers for their thoughtful feedback and engagement during the discussion period. As suggested by Reviewer **Vogk**, we provide this global summary to clearly outline (1) the key concerns raised by each reviewer, (2) the revisions we have made, and (3) explicit confirmation that all issues have been addressed in our rebuttal, even if some reviewers have not yet replied.

Our goal is to help the **ACs** evaluate the current status and post-rebuttal quality of the paper in a transparent and organized manner.

---

## Reviewer be7j — Summary of Concerns and Post-Rebuttal Resolution

**Status:** All concerns addressed in rebuttal (waiting for the reviewer's reply).

### Key Points Raised → Updates Made

1. **Fairness of baselines (HyenaDNA, TITAN, ESM-2).**
   → Added strong **DNA-LM** and PLM baselines (**HyenaDNA, ProtGPT2**) and showed they fail to maintain genome structure during autoregressive generation. Clarified why TITAN and ESM-2 cannot perform autoregressive DNA/antigen forecasting.

2. **Clarification of the “70% mismatch reduction.”**
   → Clarified that the 70% reduction is relative to the current **WHO** strain-selection system, not beth-1. Added Welch t-tests and clarified temporal splits (strict pre-2022 → post-2022, no leakage).

3. **Concern about subtype classification being “too perfect.”**
   → Added detailed biological explanation (subtypes differ by 20–40% nucleotide sequence). Provided controls: antigen-only ablations at **100% F1**; fine-tuned **HyenaDNA** at **99.81% F1**. Confirmed performance is expected and not due to leakage.

4. **Doubts about cross-subtype generalization (H7N9).**
   → Added full details: **82 test sets** used; H7N9 shows conserved evolution; clarified this is not an artifact of small sample size.

5. **Figure 1 clarity & data imbalance.**
   → Revised Fig. 1 and Appendix A.1 with explicit counts by region/year; clarified sentinel-token flow.

6. **Formatting issues.**
   → Corrected title placement, encoding artifact, and appendix/figure numbering.

7. **Concern that AntigenLM is “too small.”**
   → Added ablations showing gains come from functional-unit genome structure, not model size.

8. **Scalability.**
   → Clarified “population-scale” refers to tens of thousands of complete genomes, and explained how **AntigenLM** scales efficiently to millions.

9. **t-SNE skepticism.**
   → Added quantitative clustering metrics (Silhouette, ARI, NMI) confirming real structural separation.

10. **Regional transfer performance (H3N2-NA U.S.).**
    → Provided detailed explanation of clade shift and why NA alone appears slightly noisier while HA shifts are predicted correctly.

**Conclusion:** All concerns from Reviewer **be7j** have been fully resolved with substantial additions, clarifications, and quantitative analyses.

---

## Reviewer DjYH — Summary of Concerns and Post-Rebuttal Resolution

**Status:** All concerns addressed in rebuttal (waiting for the reviewer's reply).

### Key Points Raised → Updates Made

1. **Necessity of DNA-level modeling & need for PLM baselines.**
   → Added **ProtGPT2** and protein-only **AntigenLM** baselines. Demonstrated that PLMs cannot generate valid antigens (sentinel corruption, length drift) and lack nucleotide-level evolutionary signals.

2. **Architectural transparency.**
   → Added full architecture specification (attention heads, hidden dimension, FFN size). Updated Fig. 1B accordingly.

3. **Clarification of sentinel-token usage.**
   → Expanded Sec. 3.2 to clearly show: **no sentinel in pretraining**; used only for masked-span prediction in fine-tuning.

4. **Ambiguity in temporal conditioning (how forecasting works).**
   → Rewrote Sec. 4.3 to clarify next-month and next-season prediction setup, conditioning on prior time windows and region-specific sequences.

**Conclusion:** All comments by Reviewer **DjYH** have been fully addressed with new baselines, clearer modeling rationale, additional explanations, and expanded methodology.

---

## Reviewer Vogk and HonT — Summary and Appreciation

We sincerely thank Reviewers **Vogk** and **HonT** for their constructive engagement and for suggesting that we post this global summary. All their concerns have been fully addressed and confirmed by their responses.

---

## Final Notes for Area Chairs

Across the discussion period, we made substantial improvements:

- Added strong baselines (**HyenaDNA, ProtGPT2**, protein-only ablations).
- Added statistical significance testing and clarified evaluation procedures.
- Improved figures, appendix organization, and architectural explanations.
- Strengthened biological and temporal modeling justifications.
- Added quantitative clustering metrics, subtype-classification controls, and cross-subtype evaluations.
- Corrected all formatting/encoding issues.

All major technical, methodological, and presentation concerns raised by reviewers have been fully addressed.

---

### Author Response · Authors · 2025-11-30
**Final AC-Facing Summary**

This paper presents **AntigenLM**, a structure-aware DNA language model for influenza forecasting that explicitly preserves genome functional units during pretraining. Reviewers appreciated the novelty of applying DNA language models to viral evolution, the biologically motivated segment-aware design, strong empirical performance over evolutionary baselines, and the well-curated dataset supporting generalization across subtypes and geographies.

Early concerns focused on missing protein LM baselines, clarity on the benefits of DNA-level modeling, model architecture and methods (e.g., sentinel tokens, temporal forecasting), evaluation rigor, and overall presentation. Minor points included figure clarity, typos, and rationale for baseline choices. The authors addressed these by adding **strong baselines** (HyenaDNA, ProtGPT2, Antigen-only ablations), clarifying architecture, pretraining, fine-tuning, and evaluation procedures, improving visualizations, and fixing formatting issues. These updates make the biological motivation, model design, and practical utility of AntigenLM much clearer.

**Score changes after rebuttal (22 Nov):**
- **Reviewer DjYH:** 6 → 8  (26 Nov)
- **Reviewer be7j:** No change / no feedback
- **Reviewer HonT:** 6 → 8  (24 Nov)
- **Reviewer Vogk:** 4 → 6  (23 Nov)

Here, we list the reviewer’s comments and author’s response.

## Strengths / Contributions

**Novelty**
- First DNA LM applied to viral evolution forecasting (DjYH, HonT).
- Maintains full-genome and segment-level context, capturing both local and global co-evolutionary dependencies (be7j, Vogk).

**Applications**
- Outperforms existing evolutionary models (beth-1, LBI) in next-month/next-season prediction, rare-subtype transfer, and geographic generalization (DjYH, HonT, Vogk).
- Reduces amino-acid mismatches in HA/NA, particularly in epitope regions, which is relevant for vaccine design (Vogk, HonT).
- Generates a structured and well-curated influenza genome dataset with temporal and geographic splits, offering a useful resource to the community (DjYH, HonT).

**Language Modeling**
- Functional-unit-aware pretraining improves prediction over incomplete-genome or segment-wise variants (be7j, Vogk).
- Compact and efficient (6 layers, 384-dim), balancing model capacity and domain specificity (be7j).
- Produces interpretable embeddings and representations, supported by t-SNE and confusion-matrix analyses (Vogk).


## Weaknesses / Major Concerns

1. **Baselines & DNA vs. protein modeling** (DjYH, be7j, Vogk)
   - Originally missing strong general-purpose  protein/DNA LM comparisons; now added ProtGPT2 and HyenaDNA, showing better performance than fundation LMs  (Sec. 4.2, 5.1).

2. **Functional-unit vs. context length** (be7j)
   - Ablation studies confirm segment-level constraints are essential; long-context LMs alone don’t achieve the same performance.

3. **Evaluation rigor** (be7j, Vogk)
   - Added strict temporal splits, Welch’s t-tests, and analysis of minor subtypes (H7N9, 82 sequences). AntigenLM consistently outperforms baselines.

4. **Method clarity** (HonT, DjYH, Vogk, be7j)
   - Updated Sec. 3.4 and Fig. 1B; clarified architecture, sentinel tokens, fine-tuning, and temporal conditioning.

5. **Capacity & scalability** (be7j)
   - Small LM appropriate for influenza genomes; segment-aware batching and autoregressive training allow scaling to millions of sequences.

6. **NA epitope performance** (HonT, Vogk)
   - Explained H1N1-NA conservation; AntigenLM outperforms on H3N2-NA (Sec. 5.2).

7. **Formatting / manuscript prep** (Vogk, be7j)
   - Fixed title placement, figures, captions, and Appendix structure.

## Minor Concerns / Clarifications
- Fixed-order segment concatenation without MSA clarified (Vogk).
- Antigen-only ablations included in subtype classification (HonT).
- t-SNE visualizations supplemented with Silhouette/ARI/NMI metrics (Vogk, be7j).
- Sentinel tokens, positional encoding, dual-task loss, and boundary handling clarified (Vogk).
- Minor typos and figure issues corrected (HonT).
- Additional literature references added to contextualize novelty (Vogk).

---

### Meta-Review · Area_Chair_XsKX · 2026-01-13

**Summary:**

This paper pretrains a DNA LM on aligned genome sequences, which captures evolutionary constraints and can be finetuned for predicting antigenic variants for upcoming influenza seasons. Experiments show that the proposed method outperforms baselines using unaligned genomes.

Overall, the majority reviewers lean towards accepting this paper post rebuttal, with scores 6/6/6+/2 (one reviewer explicitly stated that they will increase to 6, and another said they would raise their score from 6). Given the reviewer opinions, the novel application, and strong results, I recommend its acceptance.

**Reviewer Concerns:**

The main concerns about formatting, baselines, and evaluation have been addressed, which prompted one reviewer to bump their score from 4 to 6.

**Reviewer Scores:**

One reviewer has stated that they will change their score from 4 to 6. The reviewer with score 2 has concerns about baselines that seem to be addressed, so that reviewer is likely to increase their score. Also, there is a reviewer who originally rated 6 who said they would raise their score. Therefore, this paper is overall a clear accept.

---

### Decision · Program_Chairs · 2026-01-26

Accept (Poster)